# Enhanced coordination interaction with multi-site binding ligands for efficient and stable perovskite solar cells

Riming Nie [1,6] ✉, Peikun Zhang[1,6], Jiaxing Gao[1,6], Cheng Wang[1,6], Weicun Chu[1], Luyao Li[2], Kaiyu Wang[3], Dongmin Qian[3], Fanrong Lin [1], Xuefeng Xia[4], Yong Wu[5], Lingfeng Chao[3], Chunyang Miao [3], Xiaoming Zhao [1], Wanlin Guo [1] ✉ & Zhuhua Zhang [1] ✉

Conventional passivating ligands bind to perovskite surfaces through only a single active site, which not only creates a resistive barrier due to dense ligand packing but also restricts the enhancement of device stability. Here, we identify an antimony chloride-N,N-dimethyl selenourea complex, $Sb(SU)_2Cl_3$, as a multi-anchoring ligand to significantly enhance perovskite crystallinity, suppress defect formation, and dramatically improve moisture resistance and overall stability. As a result, we achieve a power conversion efficiency of 25.03% in fully air-processed perovskite solar cells fabricated using a two-step method —among the highest efficiencies reported for devices prepared under ambient conditions. Remarkably, unencapsulated cells exhibited linear extrapolated $T_{80}$ lifetimes of 23,325 h during dark shelf storage. Furthermore, these unencapsulated devices demonstrate exceptional thermal and operational stability, with $T_{80}$ lifetimes of 5,004 (at 85 °C) and 5,209 hours (under 1-sun illumination), respectively, ranking them among the most stable perovskite solar cells to date.

Perovskite solar cells have become a hot topic due to their rapid increase in power conversion efficiency (PCE), and their certified efficiency has reached 26.7%[1-14]. In light of the current development of perovskite solar cells (PSCs), researchers have high expectations for their commercialization. PSCs can be fabricated by one-step or two-step fabrication methods. Compared with the one-step method, the two-step method has a commercial advantage because it does not require an anti-solvent process[8,15-18]. In the two-step method, a $PbI_2$ layer reacts with subsequently deposited organic halide salts to produce the perovskite layer. Ion interdiffusion and the formation of intermediate phases, which ultimately convert into perovskites during the thermal annealing process, determine the crystallization process[19]. Notably, controlled moisture exposure has been shown to promote intermediate hydrate phases and regulate ion diffusion kinetics, leading to improved crystallinity and film morphology[20-22]. This moisture-assisted transformation has proven crucial for achieving high-performance perovskite films under ambient fabrication conditions via the two-step method[23,24]. However, to mitigate the asynchronous crystallization caused by uncontrolled ion diffusion, most high-efficiency devices still rely on low-humidity and low-oxygen

[1]State Key Laboratory of Mechanics and Control for Aerospace Structures, Key Laboratory for Intelligent Nano Materials and Devices of the Ministry of Education, and Institute for Frontier Science, Nanjing University of Aeronautics and Astronautics, Nanjing, P. R. China. [2]School of Materials Science and Engineering, Shaanxi University of Science & Technology, Xi'an, P. R. China. [3]State Key Laboratory of Flexible Electronics (LoFE) & Institute of Advanced Materials (IAM), School of Flexible Electronics (Future Technologies), Nanjing Tech University (NanjingTech), Nanjing, China. [4]School of Electrical Engineering, Nanchang Institute of Technology, 289 Tianxiang Avenue, Nanchang, Jiangxi, China. [5]College of Mechanical and Electrical Engineering, Nanjing University of Aeronautics and Astronautics, Nanjing, P. R. China. [6]These authors contributed equally: Riming Nie, Peikun Zhang, Jiaxing Gao, Cheng Wang. ✉e-mail: rmnie@nuaa.edu.cn; wlguo@nuaa.edu.cn; chuwazhang@nuaa.edu.cn

glovebox environments, raising significant concerns regarding scalability and manufacturing cost[25,26].

In addition to crystallization control, defect passivation is equally vital for achieving highly efficient PSCs. Usually, undercoordinated $Pb^{2+}$ ions at surfaces and grain boundaries act as nonradiative recombination centers, severely limiting device PSCs[27–29]. The complexation of $Pb^{2+}$ ions is crucial for fabricating PSCs via the two-stage method fully in the atmosphere. Conventional passivation strategies, such as bulk additive incorporation and surface treatments with ammonium ligands[30,31], often introduce side effects. For example, an insulating organic layer can impede charge transport, and charge-extraction barriers have been demonstrated due to ligands in alky or aromatic spacers modified PSCs[32–34]. Furthermore, these ligands bind to the perovskite through a single active site, which leads to a resistive barrier once densely packed[8,35–40]. Although Sargent et al. passivated undercoordinated $Pb^{2+}$ at the surfaces and grain boundaries of the perovskite using dual-site-binding ligands[41], a more robust solution—finding multi-site-binding (≥3 sites) ligands—is of urgent need to simultaneously achieve deep trap passivation and low interfacial resistance for efficient charge extraction.

In this article, we develop an antimony chloride-N,N-dimethylselenourea complex, formulated as $Sb(SU)_2Cl_3$, as a multi-site passivator for $Pb^{2+}$ defects. Such a complex can bind four adjacent sites of perovskite via two Se and two Cl atoms and form an extended hydrogen-bonding network through three NH-Cl bonds and dual intramolecular/intermolecular hydrogen bonds. Detailed characterizations and analyses reveal that the $Sb(SU)_2Cl_3$ enhances crystallinity, suppresses defects, and improves charge transport across interfaces. Consequently, two-step fully air-processed PSCs achieve a champion PCE of 25.03%—among the highest reported for ambient-fabricated devices. Unencapsulated devices retain 98.98% of their initial PCE after 1584 h storage in dark conditions (20–40% RH, 25 °C), projecting a $T_{80}$ lifetime of 23,325 h, rendering them as one of the most stable PSCs to date.

## Results

### Synthesis and characterization of multi-site binding ligands

As shown in Fig. 1a, antimony chloride reacts with N,N-dimethylselenourea (SU) in dichloromethane to form a $Sb(SU)_2Cl_3$ complex, synthesized following previously reported procedures[42]. The synthesized complex is soluble in polar solvents and can form various hydrogen bonds between the amine nitrogen atoms and chloride ions. These can be categorized as either intramolecular or intermolecular hydrogen bonds, which play a critical role in promoting crystal nucleation and growth. To further probe the complex's electronic characteristics, density functional theory (DFT) calculations were conducted to generate the electrostatic potential (ESP) map (Fig. 1b). The electron-deficient (positively charged) regions are mainly localized around the amino and methyl groups, favoring the formation of hydrogen bonds with $I^-$ anions. In contrast, the chloride and selenide atoms exhibit high electron density (negatively charged regions). Efficient orbital coupling between molecular units facilitates electron transfer from donor to acceptor sites, thereby enhancing coordination or binding interactions between Cl and Se atoms within the complex and undercoordinated $Pb^{2+}$ defects in the perovskite lattice. In addition, the complex exhibits an elevated highest occupied molecular orbital (HOMO) energy level (Fig. 1b), which supports efficient hole transport. The hydrophobic methyl groups, combined with the oxygen-repelling effect of chloride ions, also contribute to improved moisture and oxidation resistance of the resulting perovskite solar cells under operational conditions.

Fourier transform infrared (FTIR) spectroscopy, ultraviolet–visible (UV–vis) spectroscopy, and X-ray diffraction (XRD) were employed to verify the formation of $Sb(SU)_2Cl_3$. As shown in Fig. 1c, the FTIR spectrum of $Sb(SU)_2Cl_3$ aligns closely with previously reported data[42]. Two broad

absorption bands at ~3300 cm$^{-1}$ and ~3200 cm$^{-1}$ correspond to N–H stretching vibrations, indicating that the *N,N*-dimethylselenourea ligands retain their hydrogen-bond donor characteristics within the complex. A strong absorption peak at 1650 cm$^{-1}$ is attributed to N–H bending, further confirming the presence of intramolecular or intermolecular hydrogen bonding. In addition, a moderate band between 1000–800 cm$^{-1}$ is assigned to C–Se stretching, supporting successful coordination of selenium with antimony. It also highlights a characteristic Se–Sb vibrational band at 350–300 cm$^{-1}$, serving as direct evidence of complex formation. XRD analysis (Fig. 1d) reveals multiple diffraction peaks between 10° and 50°, consistent with a crystalline phase. Prominent peaks at 15° and 30° suggest high structural symmetry and an ordered arrangement within the lattice of $Sb(SU)_2Cl_3$. These features underscore the periodicity and crystallinity of the complex. UV–vis absorption spectra (Fig. 1e) show distinct differences among three sample types. Pure SU in ethyl acetate (SU-EAaq) exhibits major absorption between 250–300 nm. In contrast, $Sb(SU)_2Cl_3$ in solution ($Sb(SU)_2Cl_3$-EAaq) displays enhanced intensity in the same range. Notably, the solid-state film ($Sb(SU)_2Cl_3$-S) shows a red-shift of ~10 nm and significantly increased absorption intensity. This shift is likely driven by enhanced intermolecular hydrogen bonding in the solid state, which strengthens electronic coupling. Additional characterization data are provided in Supplementary Figs. 1–3.

### Interaction between ligands and perovskite

To evaluate the interaction between the multidentate ligand $Sb(SU)_2Cl_3$ and the perovskite surface, we modeled four distinct adsorption configurations: single-site binding via Se, single-site via Cl, dual-site via Se–Cl, and quadruple-site via 2Se–2Cl (Fig. 2a–d). Charge transfer analysis at the $Sb(SU)_2Cl_3/PbI_2$-terminated surface interface revealed substantial electron accumulation (Supplementary Fig. 4), indicating enhanced interfacial bonding and defect passivation. Notably, as the number of binding sites increases, the extent of charge transfer rises while the adsorption energy decreases (Fig. 2e), suggesting stronger, more stable binding. In the most favorable configuration, Se and Cl atoms from $Sb(SU)_2Cl_3$ coordinate simultaneously with four neighboring undercoordinated $Pb^{2+}$ sites, forming four bonds within a single perovskite lattice unit. Meanwhile, the square composed of Se and Cl atoms can well match the $FAPbI_3$ lattice (Supplementary Fig. 5)[43]. This configuration exhibits the strongest charge transfer and the most stable adsorption. For clarity, pristine $FAPbI_3$ and $Sb(SU)_2Cl_3$-modified $FAPbI_3$ are hereafter referred to as the control and target samples, respectively. In addition, the terminal hydrogen atom in $Sb(SU)_2Cl_3$ forms a hydrogen bond with an iodine atom on the $PbI_2$-terminated surface, further stabilizing the interface via both chemical and hydrogen bonding. We next examined point defect formation at the $PbI_2$-terminated surface. Supplementary Fig. 6 illustrates four typical surface defects (top view), and the corresponding defect formation energies are shown in Fig. 2f. The presence of $Sb(SU)_2Cl_3$ increases the formation energies of three key defect types—iodine vacancies ($V_I$), lead vacancies ($V_{Pb}$), and anti-site defects ($I_{Pb}$)—due to strong interfacial bonding that suppresses vacancy generation and atomic substitution. Among these, the I vacancy has the lowest formation energy (0.82 eV), which is 2.34 eV lower than $V_{Pb}$, 3.06 eV lower than $I_{Pb}$, and 0.76 eV lower than the $Pb_I$ anti-site defect, consistent with prior reports identifying $V_I$ as the most common defect in perovskite films. Importantly, treatment with $Sb(SU)_2Cl_3$ in the 2Se + 2Cl binding configuration significantly increases the formation energy of $V_I$, $V_{Pb}$, and $I_{Pb}$, effectively suppressing these defects. However, the formation energy of the $Pb_I$ defect decreases slightly due to the attractive interaction between the Pb atom (substituting for an I atom) and the Cl atoms of $Sb(SU)_2Cl_3$, which facilitates the formation of $Pb_I$ anti-site defects. Notably, the presence of three relatively isolated Cl atoms in $Sb(SU)_2Cl_3$ will hold a high chance to fill the iodine vacancies for defect passivation, with a favorable binding energy of −2.03 eV, thereby enhancing the overall structural stability (Supplementary Fig. 7). Given

the diverse functional groups in $Sb(SU)_2Cl_3$, we systematically investigated its interactions across various binding sites and structural configurations. The analysis revealed that the bonding to perovskite via Cl atoms is the most favorable one (Supplementary Figs. 8–10). Comparisons between the monomeric and calixarene forms of the interacting species with the perovskite surface further confirmed that the complexation process enables stronger, multi-site binding (Supplementary Fig. 10). To probe the interaction between $Sb(SU)_2Cl_3$ and perovskite constituents ($FA^+$ and $Pb^{2+}$), we performed FTIR and nuclear magnetic resonance (NMR) spectroscopy. These analyses focused on non-covalent interactions, proton coupling, polarization, and coordination effects. The FTIR spectra (Fig. 2g) revealed enhanced N–H stretching signals in the 3600–2800 $cm^{-1}$ range, indicative of

strengthened hydrogen bonding between $Sb(SU)_2Cl_3$ and $FA^+$ ions. In addition, vibrational bands in the 1650–1500 $cm^{-1}$ range, corresponding to C=O and C=N stretching, confirmed molecular interactions between the ligand and functional groups within the perovskite lattice. In the lower frequency region (1100–900 $cm^{-1}$), spectral changes suggest the formation of Pb–N and Pb–Se coordination bonds, which likely contribute to improved charge transport and enhanced electron mobility within the perovskite. Furthermore, UV–vis absorption spectra (Fig. 2h) show distinct shifts upon the incorporation of $Sb(SU)_2Cl_3$ into the perovskite matrix, further supporting the formation of strong electronic interactions between the ligand and the perovskite. The observed red shift in absorption indicates strong interactions between $Sb(SU)_2Cl_3$ and $PbI_2$, reflecting significant

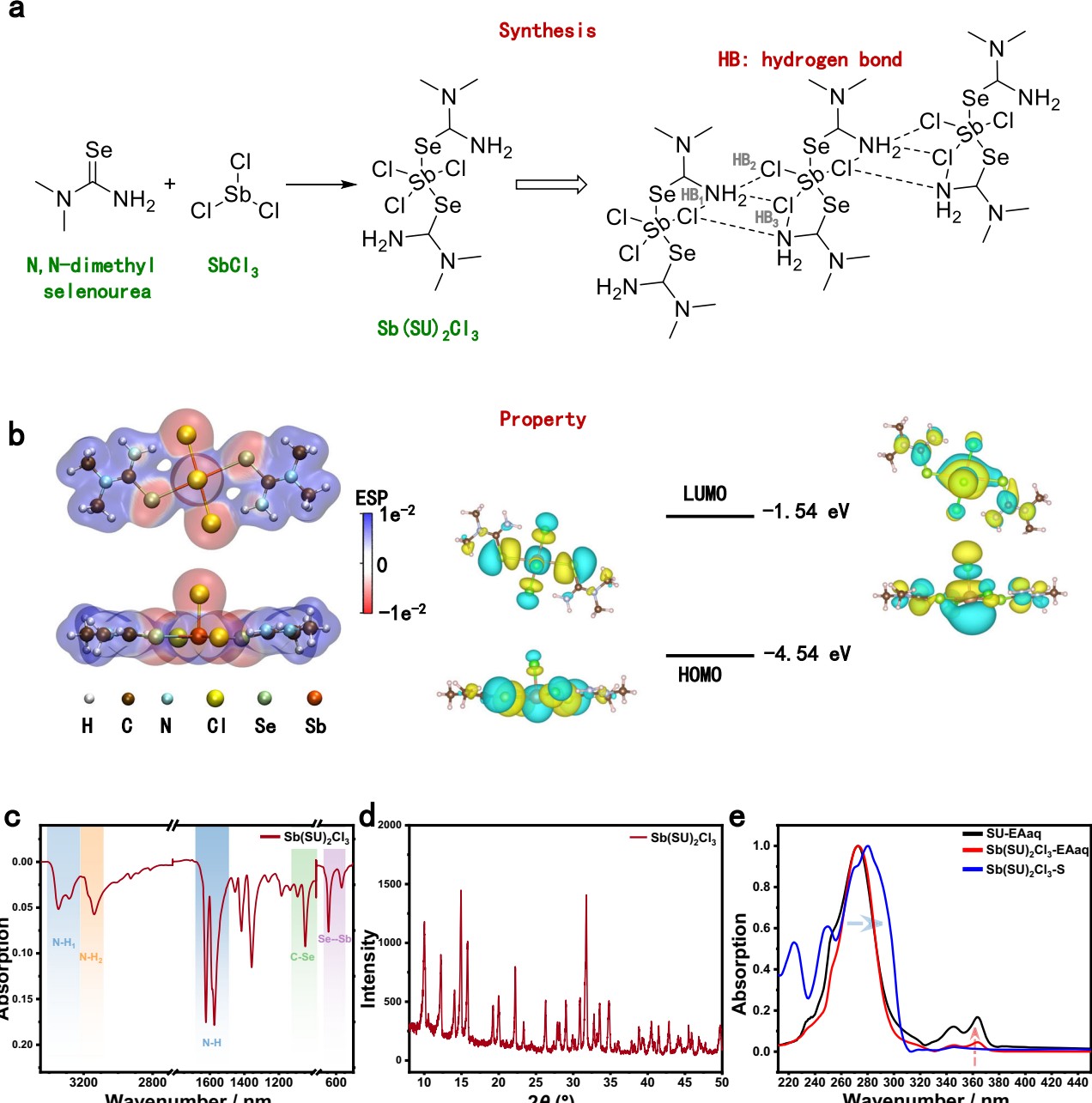

**Fig. 1 | Synthesis and characterization of $Sb(SU)_2Cl_3$. a** Schematic of the synthesis for the antimony chloride-N,N-dimethyl selenourea complex ($Sb(SU)_2Cl_3$). **b** Top view and side view for ESP and energy levels of $Sb(SU)_2Cl_3$. **c** FTIR spectrum and (**d**)

XRD pattern of $Sb(SU)_2Cl_3$. **e** UV-vis absorption spectra of pure selenourea in ethyl acetate solution (SU-EAaq), $Sb(SU)_2Cl_3$ in ethyl acetate solution ($Sb(SU)_2Cl_3$-EAaq), and $Sb(SU)_2Cl_3$ in the solid-state thin film ($Sb(SU)_2Cl_3$-S).

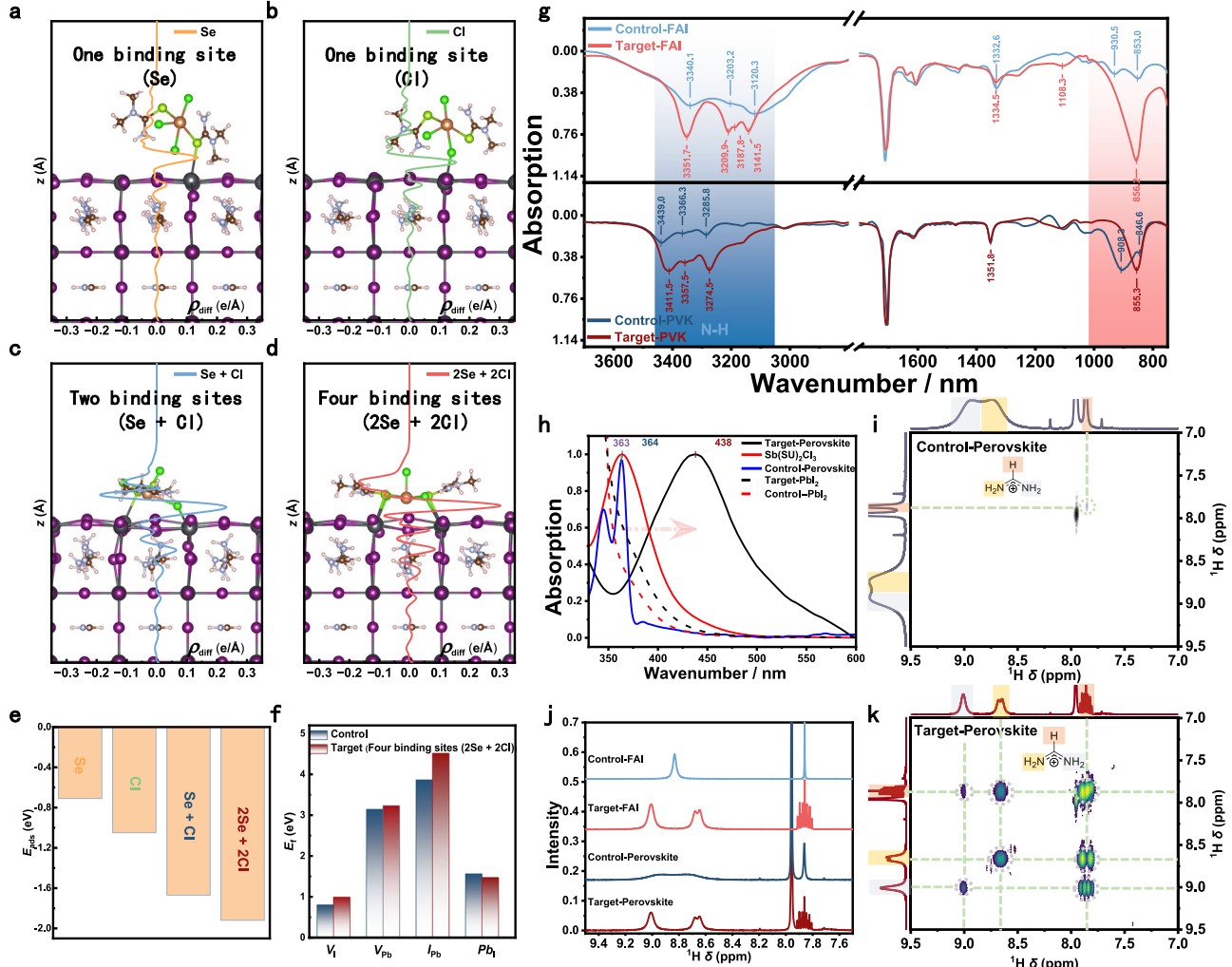

**Fig. 2 | Interaction between Sb(SU)$_2$Cl$_3$ and perovskite based on density function theory (DFT) and experimental characterization. a–d** Adsorption structures, and (**e**) adsorption energies of Sb(SU)$_2$Cl$_3$ on the surface of PbI$_2$ termination with one binding site (Se), one binding site (Cl), two binding sites (Se + Cl), and four binding sites (2Se + 2Cl). **f** Formation energies of four possible intrinsic neutral point defects for pure FAPbI$_3$ (control) and Sb(SU)$_2$Cl$_3$ modified FAPbI$_3$ with four binding sites (2Se + 2Cl) (Target). **g** FTIR spectra of FAI and perovskite without and with Sb(SU)$_2$Cl$_3$. The blue bands denote the N-H stretching peaks. **h** UV-Vis absorption spectra of FAPbI$_3$, Sb(SU)$_2$Cl$_3$, and Sb(SU)$_2$Cl$_3$ treated FAPbI$_3$. **i** $^1$H NMR spectra of FAI and perovskite without and with adsorption of Sb(SU)$_2$Cl$_3$. **j, k** 2D $^1$H-$^1$H COZY spectra of FAPbI$_3$ with and without Sb(SU)$_2$Cl$_3$, dissolved in DMSO-d6, with a small amount of DMF added to improve solubility.

changes in the material's electronic structure. This effect likely originates from coordination between the complex and Pb$^{2+}$ ions, wherein the nitrogen and selenium atoms in the selenourea ligands form Pb−N and Pb−Se bonds. These interactions modify the local chemical environment of PbI$_2$, inducing lattice distortion and reorganization, which in turn reduces the energy required for electronic transitions. This mechanism is analogous to coordination effects observed in systems such as PbI$_2$−dimethyl sulfoxide (DMSO) complexes, where solvent molecules coordinate with Pb$^{2+}$ ions, altering the Pb−I network and modulating the material's optical and electronic properties. Such interactions promote new charge-transfer transitions and confirm the formation of stable coordination complexes in solution, which benefit the subsequent crystallization and film quality of perovskite. NMR spectroscopy further reveals the impact of Sb(SU)$_2$Cl$_3$ on the chemical environments of FA$^+$ and Pb$^{2+}$ ions. In the one-dimensional $^1$H NMR spectrum (Fig. 2i), proton coupling and polarization effects induced by the complex result in an upfield shift and splitting of the FA$^+$ proton signal (originally at 8.83 ppm). This shift is also observed in the perovskite precursor solution, indicating strong hydrogen bonding between FA$^+$ and the complex, which enhances structural stability. Two-dimensional $^1$H−$^1$H COZY spectra (Fig. 2j, k) show increased

coupling between FA$^+$ protons, further confirming that the complex reduces defect states and stabilizes the ionic environment within the perovskite lattice. Raw NMR data are provided in Supplementary Figs. 11–16. Taken together, FTIR and NMR results demonstrate that the multidentate ligand Sb(SU)$_2$Cl$_3$ enhances both the structural stability and optoelectronic properties of perovskite materials. These improvements are achieved through a combination of mechanisms, including strengthened hydrogen bonding, proton polarization, and Pb$^{2+}$ coordination, which jointly optimize charge transport and light absorption. Comparative studies of SbCl$_3$, SU, and the Sb(SU)$_2$Cl$_3$ complex with the perovskite precursor (Supplementary Figs. 17 and 18) revealed that Sb(SU)$_2$Cl$_3$ exhibits the strongest binding affinity. Importantly, the complex is not a simple physical mixture of SbCl$_3$ and SU, but rather a chemically integrated structure in which halide and SU ligands synergistically stabilize the PbI$_2$ coordination environment and enhance interactions with FA$^+$ ions. The detailed explanation can be found in Supplementary Figs. 17 and 18 of the Supporting Information.

To investigate the influence of the multidentate ligand Sb(SU)$_2$Cl$_3$ on the nucleation and growth of perovskite crystals, we fabricated perovskite films using a two-step method. In the control

process (Supplementary Fig. 19a), a $PbI_2$ solution in DMF/DMSO was spin-coated onto the substrate, followed by spin-coating of an FAI solution in IPA. The films were then annealed at 150 °C for 10 min to form perovskite layers. In the modified process (Supplementary Fig. 19b), a defined amount of $Sb(SU)_2Cl_3$ was added to the $PbI_2$ solution, while the remaining steps were identical to those of the control. Surface morphology and crystallographic structure of the resulting films—prepared under ambient conditions (20–40% relative humidity)—were characterized using scanning electron microscopy (SEM) and X-ray diffraction (XRD) (Supplementary Figs. 20 and 21). Compared to the control, the target films displayed a denser and more uniform morphology, accompanied by diminished $PbI_2$ diffraction peaks and a reduced full width at half maximum (FWHM), indicating enhanced film quality and improved crystallinity resulting from the incorporation of $Sb(SU)_2Cl_3$. The dependence of film morphology and crystallinity on $Sb(SU)_2Cl_3$ concentration is presented in Supplementary Figs. 22 and 23. To further assess crystallization dynamics, in situ UV–vis absorption spectroscopy was performed during the two-step film formation (Fig. 3a, b). The crystallization of the target film proceeded more slowly during thermal annealing, with the transition to the α-$FAPbI_3$ phase completed at 2.98 s, in contrast to 1.29 s for the control film. This delayed and more ordered crystallization process, driven by interactions between $Sb(SU)_2Cl_3$ and $FAPbI_3$, led to enhanced film crystallinity and suppressed defect formation. In situ grazing-incidence wide-angle X-ray scattering (GIWAXS) measurements (Supplementary Fig. 24) confirmed the evolution of the α-$FAPbI_3$ phase, with a characteristic diffraction signal appearing at $q \approx 1.0$ Å$^{-1}$. The α-phase emerged more immediately in the target film compared to the control, suggesting that $Sb(SU)_2Cl_3$ accelerates the nucleation of the desired perovskite phase, thereby improving overall film quality.

To visualize the distribution of residual strain within perovskite films, grazing-incidence X-ray diffraction (GIXRD) was performed (Fig. 3c, d), revealing depth-dependent variations in residual stress and microstrain along the (012) plane. Upon incorporation of $Sb(SU)_2Cl_3$, the absolute residual stress decreased from 16.3 MPa (control) to 8.6 MPa (target), indicating partial stress relaxation in the perovskite lattice (Supplementary Fig. 25). X-ray photoelectron spectroscopy (XPS) was conducted to examine the chemical states of lead (Fig. 3e, f). The signal corresponding to metallic $Pb^0$ was significantly suppressed, and the Pb 4$f$ binding energy exhibited a + 0.25 eV shift upon incorporation of $Sb(SU)_2Cl_3$. These observations indicate reduced residual $PbI_2$ and more complete conversion to the perovskite phase, as well as strong coordination interactions between $Pb^{2+}$ and the introduced ligands. To further investigate the effects on crystallinity and crystal orientation, grazing-incidence wide-angle X-ray scattering (GIWAXS) was employed. The 2D GIWAXS patterns at an incident angle of 0.800° (Fig. 3g, h) show diffraction signals corresponding to α-$FAPbI_3$, δ-$FAPbI_3$, and $PbI_2$ phases in both control and target films. After $Sb(SU)_2Cl_3$ incorporation, the intensities of δ-phase and $PbI_2$ peaks decreased markedly, while the α-phase exhibited stronger, more uniform orientation, indicating enhanced phase purity and improved crystallinity. To quantify crystallographic orientation, the azimuthal angle ($\chi$) dependence of the integrated diffraction intensity was analyzed. The orientation fraction was derived using the ratio $A_{xy}/(A_{xy} + A_z)$, representing the proportion of face-on to edge-on crystallites. Pole figure analysis of the (110) diffraction from α-$FAPbI_3$ (Fig. 3i, j) showed an increased out-of-plane orientation in the target film (53.06%) compared to the control (43.40%), suggesting that $Sb(SU)_2Cl_3$ promotes more vertically aligned crystal domains. In addition, the variation in the out-of-plane diffraction peak with incidence angle reflects changes in $d$-spacing from the surface to the bulk of the film. The $Sb(SU)_2Cl_3$-modified films exhibited a greater $d$-spacing gradient (Fig. 3k, l), which may arise from the ligand-induced lattice modulation and preferential out-of-plane crystal growth.

## Effect of ligands on the device performance

PSCs with the device structure FTO/$SnO_2$/perovskite/phenethy-lammonium iodide (PEAI)/spiro-OMeTAD/Au were fabricated to investigate the effect of multi-site binding ligands ($Sb(SU)_2Cl_3$) on device performance (Supplementary Fig. 26). Figure 4a shows the current density–voltage ($J$–$V$) curves of devices with and without the incorporation of $Sb(SU)_2Cl_3$. The control device exhibited a short-circuit current density ($J_{SC}$) of 24.94 mA cm$^{-2}$, an open-circuit voltage ($V_{OC}$) of 1.14 V, and a fill factor (FF) of 81.6%, corresponding to a power conversion efficiency (PCE) of 23.19%. Upon introducing the multi-site binding ligands, $J_{SC}$, $V_{OC}$, and FF increased to 25.66 mA cm$^{-2}$, 1.18 V, and 82.7%, respectively, yielding a PCE of 25.03%. The modified device also showed a steady-state PCE of 24.83% (inset of Fig. 4a), consistent with the value obtained from the $J$–$V$ measurement. The corresponding external quantum efficiency (EQE) spectra of the PSCs are shown in Fig. 4b. By integrating the overlap between the EQE spectra and the standard AM 1.5 G solar photon flux, current densities of 24.43 mA cm$^{-2}$ and 24.81 mA cm$^{-2}$ were obtained for the control and target devices, respectively. These values are in good agreement with the $J_{SC}$ values measured from the $J$–$V$ curves. The champion device was certified at an accredited photovoltaic certification laboratory (Shanghai Institute of Microsystem and Information Technology, Chinese Academy of Sciences (SIMIT)), confirming the reliability of the PCE (Supplementary Fig. 27). A reduced hysteresis was also observed after introducing $Sb(SU)_2Cl_3$ (Supplementary Fig. 28). The additive is also effective in the one-step fabrication process (Supplementary Fig. 29). Compared with the control cells, the target cells exhibited a more concentrated performance distribution (Supplementary Fig. 30). To investigate the origin of the enhanced device performance, various characterizations were conducted. The short-circuit current density ($J_{SC}$) is jointly determined by light-harvesting efficiency (LHE), electron injection yield, and charge collection efficiency. Comparable LHE was confirmed by the nearly identical absorption spectra (Fig. 4c). Nyquist plots obtained from electrochemical impedance spectroscopy (EIS) revealed reduced series resistance ($R_s$) and transport resistance ($R_{trans}$), along with an increased recombination resistance ($R_{rec}$), indicating suppressed charge recombination and enhanced charge transfer efficiency (i.e., improved electron injection yield and charge collection efficiency) upon incorporation of $Sb(SU)_2Cl_3$ (Fig. 4d).

The fitting parameters based on the equivalent circuit shown in the inset are summarized in Supplementary Table 1. Charge transport within the hole transport material (HTM) and at the perovskite/HTM interface is represented by the semicircles in the high- and low-frequency regions, respectively[44]. The incorporation of $Sb(SU)_2Cl_3$ improves perovskite film quality and reduces defect density, as confirmed by the increased photoluminescence (PL) intensity (Fig. 4e). To further investigate carrier dynamics, time-resolved photoluminescence (TRPL) spectroscopy was performed, with the resulting spectra fitted using a biexponential function (Fig. 4f)[45].

$$y = A_1 * \exp^{(-x/t_1)} + A_2 * \exp^{(-x/t_2)} + y_0 \tag{1}$$

where the longer decay time ($t_1$) corresponds to radiative recombination, and the shorter decay time ($t_2$) corresponds to defect-induced nonradiative recombination[46,47]. The significantly reduced defect-assisted recombination was evidenced by the increased carrier lifetime after incorporation of $Sb(SU)_2Cl_3$. Temperature-dependent current–voltage measurements were used to determine the average activation energy of trapped carriers, calculated using

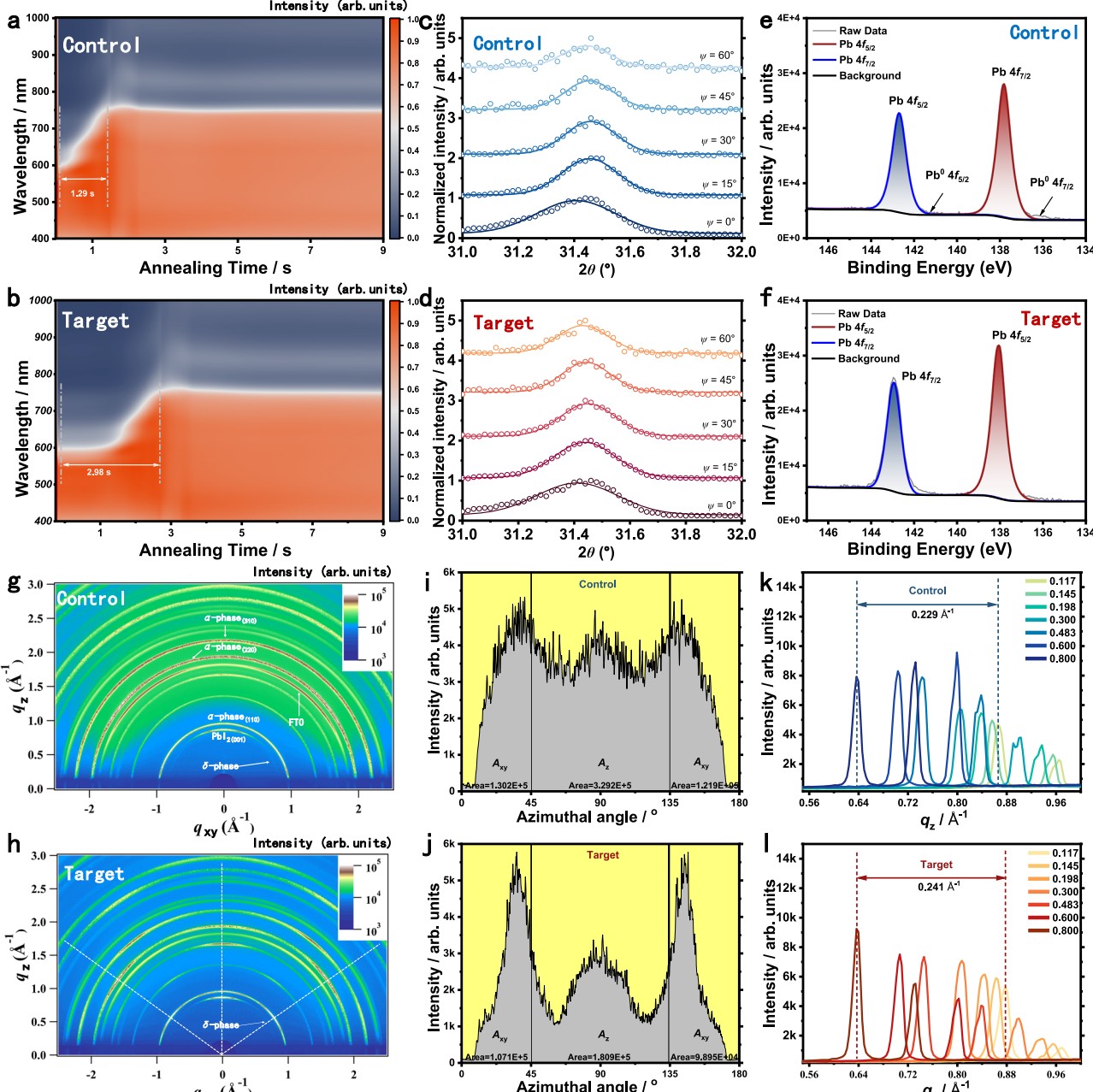

**Fig. 3 | Effect of multi-site binding ligands (Sb(SU)₂Cl₃) on the perovskite layers.** **a**, **b** In situ UV-Vis absorption spectroscopy for control (**a**) and target (**b**) groups, (**c**, **d**) Grazing-incidence X-ray diffraction (GIXRD) profiles of perovskite thin films (012) crystallographic plane. High-resolution Pb 4$f$ XPS peaks of the control (**e**) and target (**f**) perovskite film. **g**, **h** GIWAXS images of the control and target samples at an incidence angle of 0.8°. **i**, **j** Pole figure plots from the (100) lamellar diffraction as a function of incidence angles of the control and target samples. **k**, **l** Corresponding out-of-plane line cuts of the GIWAXS images as a function of incidence angles of the control and target samples.

the Richardson–Dushman equation[48]:

$$J \propto e^{-\Delta E/kT} \qquad (2)$$

where $\Delta E$, $k$, and $T$ represent the electron activation energy, Boltzmann constant, and absolute temperature, respectively. Compared with the control sample, the target sample exhibited a lower trap activation energy, indicating shallower traps in the perovskite films upon addition of the multi-site binding ligand Sb(SU)₂Cl₃ (Fig. 4g and Supplementary Fig. 31). The thermally stimulated current (TSC) curves and space-charge-limited current (SCLC) measurements further confirmed a reduction in defect density (Fig. 4h, i). Therefore, we conclude

that the incorporation of Sb(SU)₂Cl₃ enhances the crystalline quality and reduces defect density, leading to improved device performance.

Time-resolved X-ray diffraction (XRD) measurements were performed at 110 °C and 70% relative humidity (R.H.) to assess the phase stability of the control and target samples (Fig. 5a, b). Compared with the control sample, the target sample exhibited a slower degradation rate. In the fresh control sample, a relatively strong PbI₂ diffraction peak—attributed to fabrication under ambient conditions (20–40% R.H.)—was significantly suppressed after incorporating Sb(SU)₂Cl₃. Unencapsulated devices were stored in ambient air (20–40% R.H.) to evaluate dark shelf stability. The target cell retained 98.98% of its initial PCE after 1584 h of storage, corresponding to a projected $T_{80}$ lifetime

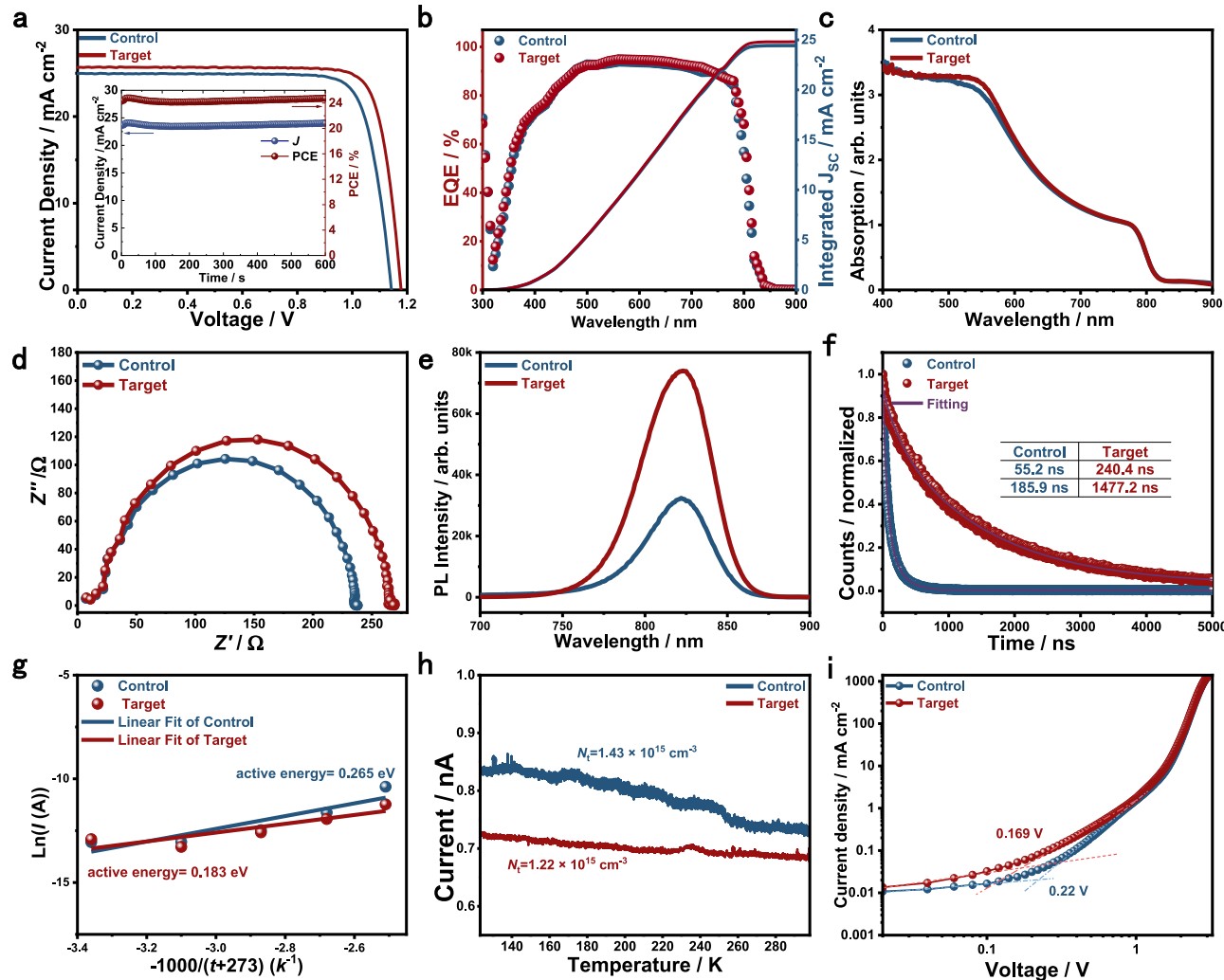

**Fig. 4 | Device performance. a** $J$–$V$ characteristics under standard illumination conditions (100 mW cm⁻², AM 1.5 G) and the stabilized PCE (the inset), (**b**) EQE spectra, (**c**) UV-vis absorption spectra, (**d**) Nyquist plots under dark condition, (**e**) steady-state photoluminescence spectra, and (**f**) carrier lifetime of the control and target samples, (**g**) Dependence of dark current on temperature of the control and target perovskite solar cells, (**h**) Thermally stimulated current (TSC) spectra of the control and target samples, (**i**) Space charge limiting current (SCLC) curves of the control and target samples.

of 23,325 h by linear extrapolation, whereas the control cell's PCE decreased to 80.74% after 1008 h (Fig. 5c, d).

Thermal stability was tested by storing the devices at 85°C following the ISOS-D-2 protocol. While Kim et al. previously provided mechanistic insights into de-doping effects[49], we employed Poly [bis(4-phenyl)(2,4,6-trimethylphenyl)amine] (PTAA) as the hole transport layer (HTL) to better isolate the intrinsic thermal stability of the perovskite layer. Under these conditions, the target cells maintained 91.46% of their initial PCE after 1012 h of storage, corresponding to a projected $T_{80}$ lifetime of 5004 h, whereas the control cells retained only 76.73% of their initial PCE after 528 h (Fig. 5e, f). Device operational stability was evaluated by maximum power point tracking (MPPT). The target cell retained 99.98% of its initial PCE after 969 h, corresponding to a projected $T_{80}$ lifetime of 5209 h (Fig. 5g, h), while the control cell retained 55.67% after 489 h. These results demonstrate that the device is among the most stable and efficient perovskite solar cells fabricated under ambient conditions reported to date (Supplementary Tables 2–4). To elucidate the origin of the enhanced stability, first-principles calculations were performed to evaluate the adsorption energies of $O_2$ and $H_2O$ molecules on the FAPbI₃ (100) surface without and with Sb(SU)₂Cl₃ (Supplementary Figs. 32 and 33; Supplementary Tables 5 and 6). The increased adsorption energies of $O_2$ and $H_2O$

indicate significantly improved resistance to moisture upon addition of Sb(SU)₂Cl₃, consistent with the observed enhancements in thermal, ambient, and operational stability.

## Discussion

This study was initiated with the understanding that most ammonium-based ligand additives and surface passivators interact with the perovskite lattice through a single active binding site, often resulting in dense ligand packing that introduces unwanted resistive barriers. We hypothesized that this limitation could be overcome by employing ligands capable of binding at multiple active sites on the perovskite surface. Such multi-site coordination with undercoordinated $Pb^{2+}$ ions effectively suppresses defect formation and promotes enhanced crystallinity. In particular, the Sb(SU)₂Cl₃ complex, which forms both chemical and hydrogen bonds with the perovskite lattice, significantly improves structural stability and moisture resistance. Perovskite solar cells fabricated entirely in ambient air using a two-step method and incorporating these multi-site binding ligands achieved a power conversion efficiency (PCE) of 25.03%—among the highest reported for air-processed devices. These devices also exhibited exceptional long-term stability, retaining approximately 99% of their initial performance after 1584 h of storage in ambient air (20–40% relative humidity), with a

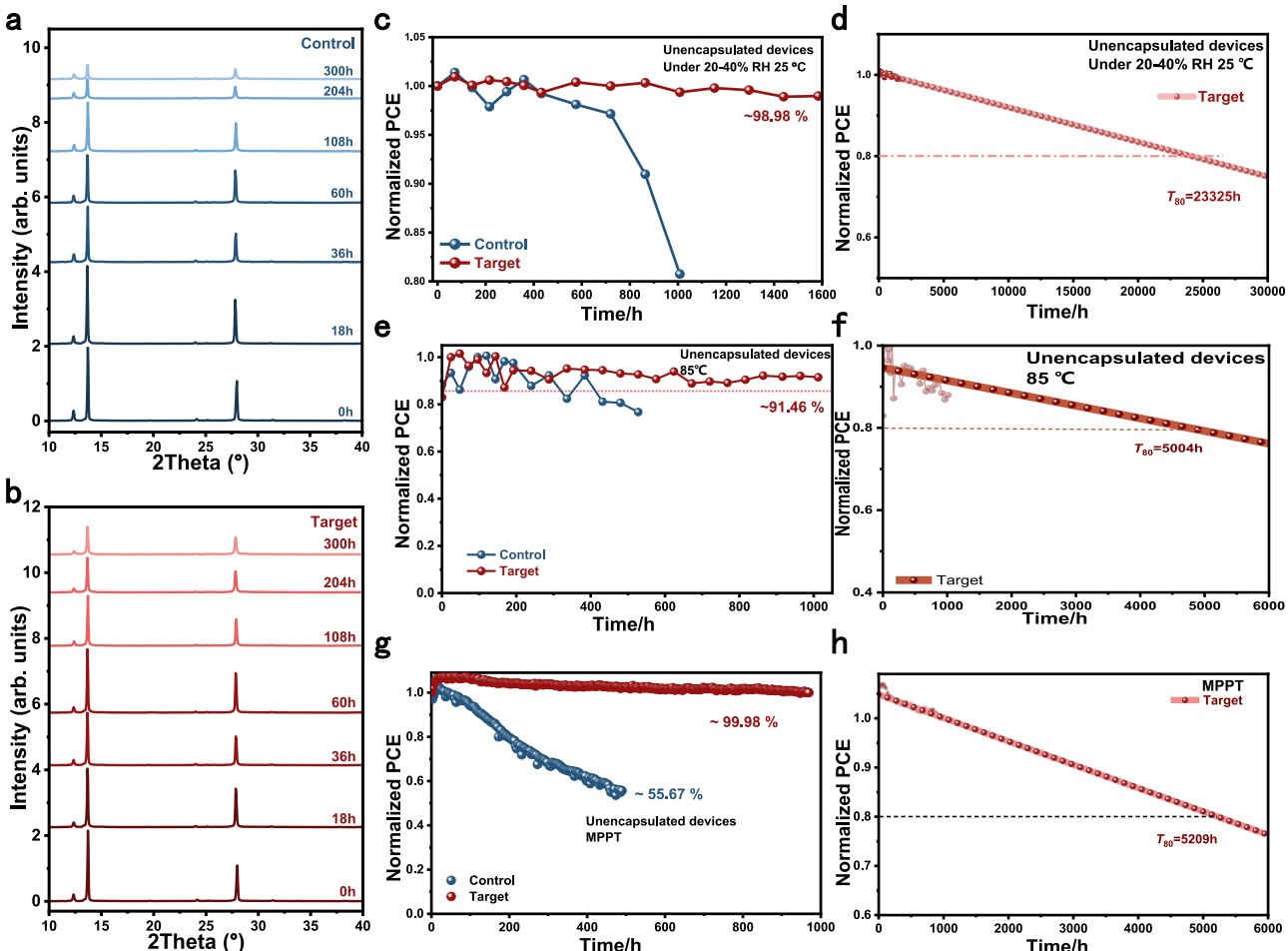

**Fig. 5 | Phase stability and device stability. a**, **b** XRD patterns of control and target perovskite films aged in ambient conditions (70% humidity) and at 110 °C. The samples were prepared in the atmosphere (20–40% humidity) and at room temperature. **c**, **d** Dark shelf stability of unencapsulated control and target PSCs aged in ambient conditions at room temperature with R.H. around 20–40% and the corresponding linear extrapolation. **e**, **f** Evolution of the PCEs tracked under continuous heating at about 85 °C following the ISOS-D-2 protocol and the corresponding linear extrapolation. **g**, **h** Evolution of the PCEs tracked under continuous one sun light soaking under the maximum power point and the corresponding linear extrapolation.

projected $T_{80}$ of 23,325 h under dark shelf conditions. This work highlights the potential of multi-site binding ligands as a promising strategy for simultaneously enhancing both the efficiency and durability of perovskite solar cells.

## Methods

### Synthesis of the Sb(SU)₂Cl₃ complex

The Sb(SU)$_2$Cl$_3$ complex was synthesized following a previously reported procedure[42]. Briefly, 200 mg (0.877 mmol) of SbCl$_3$ was dissolved in 60 mL of dichloromethane to form solution A. Separately, 265 mg (1.754 mmol) of N,N-dimethyl selenourea was dissolved in 60 mL of dichloromethane to form solution B. Solution A was added dropwise to solution B over 30 minutes, and the resulting mixture was stirred under a nitrogen-protected atmosphere for 2 h. A yellow precipitate formed, which was collected by filtration through Whatman filter paper and washed several times with dichloromethane to yield the final product, the Sb(SU)$_2$Cl$_3$ complex. The yield was 95%. $^1$H NMR (400 MHz, DMSO-d6) δ 7.69 (2H), 3.18 (6H). $^{13}$C NMR (400 MHz, DMSO-d6) δ 177.42, 55.38.

### Device fabrication

The SnO$_2$ solution was diluted with deionized water at a volume ratio of 1:3. The diluted solution was spin-coated onto FTO glass at 3000 rpm for 30 s, followed by annealing at 150 °C for 30 min. To

prepare the perovskite film, 691.5 mg (1.5 M) of PbI$_2$ was dissolved in a mixed solvent of DMF (0.9 mL) and DMSO (0.1 mL). The resulting solution was spin-coated onto the substrate at 1500 rpm for 30 s and annealed at 70 °C for 60 s. The PbI$_2$ film was then cooled at room temperature for 60 s. For the modified samples, a Sb(SU)$_2$Cl$_3$ solution was prepared by dissolving 0.5 mg of Sb(SU)$_2$Cl$_3$ in 1 mL of DMF. Various volumes (0, 0.5, 1, and 2 µL) of the Sb(SU)$_2$Cl$_3$ solution were added to 100 µL of the PbI$_2$ solution, respectively. Separately, 90 mg of FAI, 9 mg of MACl, and 6.39 mg of MAI were dissolved in 1 mL of IPA to form the organic halide salt solution. This solution was spin-coated onto the PbI$_2$ film at 2000 rpm for 30 s. Finally, the films were annealed at 150 °C for 15 min to complete the perovskite film formation. For surface passivation, 4.98 mg of PEAI was dissolved in 1 mL of IPA and spin-coated onto the samples at 5000 rpm for 30 s. To prepare the hole transport layer solution, 260 mg of bis(trifluoromethane)sulfonimide lithium salt (LiTFSI) was dissolved in 1 mL of acetonitrile to form the lithium salt solution, and 375 mg of FK209 Co(III) TFSI salt (Lumtec) was dissolved in 1 mL of acetonitrile to form the cobalt salt solution. Subsequently, 72.3 mg of spiro-OMeTAD, 35 µL of the lithium salt solution, 30 µL of 4-tert-butyl-pyridine, and the cobalt salt solution were dissolved in 1 mL of chlorobenzene. The hole transport layer solution was spin-coated at 3000 rpm for 30 s. A 70 nm gold layer was thermally evaporated onto the samples to form the metal electrode.

## Stability testing

For maximum power point tracking (MPPT) measurements, the PEAI layer was omitted[50]. A hole transport layer solution was prepared by dissolving 72.3 mg of Spiro-OMeTAD, 7 mg of PTAA, 35 μL of Li-TFSI, and 28.8 μL of 4-tert-butylpyridine in 1 mL of chlorobenzene. This solution was spin-coated onto the samples at 3000 rpm for 20 s. For thermal stability tests, 20 mg of PTAA and 2.25 mg of DPI-TPFB were dissolved in 1 mL of chlorobenzene to form the hole transport layer solution. This solution was spin-coated at 1500 rpm for 30 s and annealed at 70 °C for 5 min[51]. All other procedures followed the standard device fabrication process described in the Device Fabrication section.

## Material and device characterization

An X-ray diffractometer (D/MAX2500V/PC, Rigaku, Japan), UV–vis spectrophotometer (Jasco V-780), field-emission scanning electron microscope (Hitachi S-4800, Hitachi High-Technologies), and Thermo Scientific Flash 2000 analyzer were used to characterize the fundamental properties of the materials. A solar simulator (Newport, Oriel Class A, 91195 A) coupled with a source meter (Keithley 2400) was used to measure the J–V curves. The light intensity was calibrated using a standard reference cell. An internal quantum efficiency system (Oriel, IQE 200B) was used to measure the EQE spectra. Electrochemical impedance spectroscopy (EIS) was performed using an AUTOLAB (AUT302N) system under a bias voltage of 0.9 V and a frequency range from 100 kHz to 0.1 Hz. A FluoTime 300 spectrometer (PicoQuant) was used to acquire steady-state and time-resolved photoluminescence (PL) spectra. A Fourier transform infrared spectrometer (Nicolet iS50) and a 400 MHz NMR spectrometer (Bruker) were used to measure FTIR and NMR spectra, respectively. GIWAXS measurements were carried out at the Pohang Accelerator Laboratory in South Korea.

## Theoretical calculations

Vienna *ab*-initio simulation package (VASP) was used to perform the density function theory (DFT) calculations[52–54]. The exchange-correlation functional was carried out using the Perdew-Burke-Ernzerhof (PBE) with generalized gradient approximation (GGA)[55]. The $FAPbI_3$ (100) surfaces were modeled using $3 \times 3$ slabs. A plane wave basis with the cutoff energy of 400 eV was chosen in this work. A $3 \times 3 \times 1$ k-point mesh was used in these calculations. We consider the van der Waals (vdW) interaction using the DFT-D3 method[56]. A vacuum layer along the out-plane direction of 15 Å was constructed to restrain the interactions between adjacent slabs. The residual force and energy convergence thresholds were set to 0.02 eV Å$^{-1}$ and $10^{-4}$ eV, respectively.

The definition formula of charge density difference is:

$$\Delta\rho = \rho\text{total} - \rho FAPbI_3 - \rho mol \quad (3)$$

Where $\rho_{total}$ is the total charge density of the adsorption interface, $\rho_{FAPbI3}$ and $\rho_{mol}$ are the charge densities of the $FAPbI_3$ (100) surface and the molecular, respectively.

The adsorption energy is defined as:

$$E_{ads} = E_{total} - E_{FAPbI_3} - E_{O_2/H_2O} \quad (4)$$

Where $E_{total}$ is the total energy of the $O_2/H_2O$ adsorption interface, $E_{FAPbI_3}$ and $E_{O2/H2O}$ are the energies of $FAPbI_3$ (100) surface and $O_2/H_2O$, respectively.

## Reporting summary

Further information on research design is available in the Nature Portfolio Reporting Summary linked to this article.

## Data availability

The data that supports the findings of the study are included in the main text and supplementary information files or upon request from the corresponding authors. Source data are provided in this paper.

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

## Acknowledgements

R.N. acknowledges support from the National Natural Science Foundation of China (52203359) and the supporting funds for talents of Nanjing University of Aeronautics and Astronautics. W.G. acknowledges support from the National Natural Science Foundation of China (T2293691), Natural Science Foundation of Jiangsu Province (BK20243065), National Key Research and Development Program of China (2019YFA0705400), the Fundamental Research Funds for the Central Universities (NJ2024001, NC2023001, NJ2023002) and the Fund of Prospective Layout of Scientific Research for NUAA (Nanjing University of Aeronautics and Astronautics). Z.Z. Acknowledges support from the National Natural Science Foundation of China (12225205, U2441272), National Key Research and Development Program of China (2024YFA1409600), Jiangsu Province NSF (BK20243044). The authors acknowledge the Center for Microscopy and Analysis of NUAA for characterization support.

## Author contributions

R.N. conceived the idea, designed and performed the experiments, and wrote the manuscript with Z.Z.; R.N., W.G., and Z.Z. supervised the project. P.Z. carried out theoretical calculations. J.G. and L.L. carried out the stability tests. C.W. carried out the synthesis and characterization. W.C. fabricated the solar cells. X.Z. and Y.W. provided help with the manuscript writing. K.W. and L.C. measured and analyzed the in situ grazing-incidence wide-angle X-ray scattering (GIWAXS) measurements. D.Q. and C.M. measured and analyzed the in situ UV–vis absorption spectra. F.L. measured and analyzed the thermally stimulated current (TSC) spectra. X.X. guides the device fabrications. All authors discussed, commented on, and revised the manuscript.

## Competing interests

The authors declare no competing interests.
