## [Transparent Peer Review file · Nature Communications]

Enhanced Coordination Interaction with Multi-Site Binding Ligands for Efficient and Stable Perovskite Solar Cells

Corresponding Author: Professor Riming Nie

Version 0:

Reviewer comments:

Reviewer #1

(Remarks to the Author)

This manuscript presents an interesting approach to improving the efficiency and stability of perovskite solar cells (PSCs) by introducing multi-site binding ligands ($\text{Sb}(\text{SU})_2\text{Cl}_3$). The efficiency of the device is impressive for a two-step perovskite solar cell, and the stability is promising. However, there are areas where clarification and additional experimentation would strengthen the manuscript:

1. The authors emphasize the benefits of the multi-site binding additive ($\text{Sb}(\text{SU})_2\text{Cl}_3$) for two-step fabrication, yet the rationale for its specificity to this method is not entirely clear. Did the authors attempt to incorporate this additive in one-step fabrication? Demonstrating its performance in one-step devices would help confirm whether the additive is inherently advantageous for two-step processing or if its effects are more general.
2. Relatedly, manuscript does not provide a direct explanation of how $\text{Sb}(\text{SU})_2\text{Cl}_3$ specifically addresses challenges unique to the two-step process. For example, how does the additive influence the interaction between FAI and PbI_2 or mitigate asynchronous crystallization dynamics? The connection between the additive's functionality and the intrinsic characteristics of two-step fabrication should be explicitly discussed.
3. One area where the manuscript could be significantly strengthened is through the inclusion of dynamic or in-situ experiments to directly investigate the additive's role in crystallization kinetics and film formation. Techniques such as in-situ XRD or GIWAXS would offer valuable insights into the phase evolution during the fabrication process, while in-situ PL or TRPL could help track defect dynamics and charge carrier behavior during annealing. Incorporating such experiments would provide robust, real-time evidence to support the proposed mechanisms of improved film quality and enhanced crystallinity induced by $\text{Sb}(\text{SU})_2\text{Cl}_3$, further solidifying the additive's role in achieving these improvements.
4. The manuscript could be enhanced by explicitly comparing the stability and efficiency metrics of devices incorporating $\text{Sb}(\text{SU})_2\text{Cl}_3$ to other state-of-the-art PSCs fabricated via two-step and one-step methods, especially the MPPT stability at $>65^\circ\text{C}$.
5. There is a discrepancy in the description of Figure 6g and 6h. The legend indicates MPPT (maximum power point tracking), but the text describes the condition as open-circuit. This inconsistency should be corrected for clarity and accuracy.
6. I also recommend shorten the manuscript to 4-5 figures.

Reviewer #2

(Remarks to the Author)

In this manuscript, the author incorporated a ligand (a complex of antimony chloride-N,N-dimethyl selenourea, $\text{Sb}(\text{SU})_2\text{Cl}_3$) with multi-binding-site into PbI_2 precursor in a two-step spin-coating method, thus make it to be an additive of perovskite. It is demonstrated that this multi-site binding ligands can suppress the formation of various defects, enhances the crystallinity of perovskites, and greatly increases the stability of the structure and the ability to resist water, thus attributing to a PCE of 25.03% for air-fabricated PSCs with enhanced stability. Overall, the author proposed a new additive through complexing the antimony chloride and N,N-dimethylselenene (SU), however the concept of Multi-Site binding with perovskite has been variously reported, and some direct support experiments and analyzation need to supplemented. In addition, the PCE of 25.03% is difficult to support it among the highest values of reported air-fabricated devices. At this stage, it is difficult to recommend the acceptance. Here are the detailed comments.

#1: In the part of introduction: (1) the description of the "three stages in a two-step spin-coating" needs supporting reference, I wonder is there a second stage of involving incorporating water into the reactants, since if we fabricate the perovskite film in

glove-box, including the annealing, there is no water. (2) The first paragraph of introduction mainly describes the Asynchronous crystallization, the second paragraph of introduction focus on the undercoordinated Pb²⁺ ions and its corresponding treatment, the logical interaction between these two paragraphs need to be modified and supplemented.

#2: There experimental data proved that the interaction between Sb(SU)₂Cl₃ and perovskite can also be Pb-N and hydrogen bond, why these bindings have not been included in the DFT calculation?

#3: Considering there are various binding approaches (one site, two site four site, maybe three site, Pb-N, Pb-Se, Pb-Cl and hydrogen bond), it would be better and meaningful to provide a discussion about the practical interaction method between perovskite and additives, or which binding approach the author think is most reasonable and possesses more possibilities.

#4: Considering the core idea of this work is enhanced coordination interaction, the direct comparative experiment is suggested to be supplemented to prove that the complexing can enhance the binding energy.

#5: In Figure 3, the main data focus on the chemical states changes of FA⁺ (Supporting the interaction between FAI and additives), however, the DFT results in Figure 2 reveal that the all four binding sites focus on the Pb²⁺. The experiments that can proved the enhanced coordination between additive with Pb²⁺ is suggested to be supplemented. In addition, the author mentioned that "NMR spectroscopy further highlighted significant effects of the complex on the chemical environments of FA⁺ and Pb²⁺ ions.", however in the following discussion there no discussion of the chemical environments of Pb²⁺ ions. The peak position and peak shift are suggested to be marked in the Figure 3a.

#6: In Figure 4a-4d, the denser morphology and lower Pbl₂ peaks can not support the conclusion of improved crystallinity, the FWHM should been analyzed. What the reason for the optimized crystallization? especially considering the ambient air condition. The in-situ measurement, is suggested to performed to reveal the influence of additive on perovskite crystallization.

#7: Considering the main effects of the additive is reducing the defects in perovskite film, the direct experiments to demonstrate that the reduced defects should been supplemented, such as SCLC, thermal admittance spectroscopy....

#8: In Figure 5, The parameter statistics of control PSCs under reverse and forward scan should been provided. Photovoltaic parameter statistics of multiple devices need to be supplemented. In addition, among the highest efficiencies is a too broad description, if the author wants to evaluate this efficiency in this way, the efficiency statistic of the reported works should be provided to make a direct comparison. Besides, the certificated PCE should also been provided.

#9: Is there any surface passivation? In Figure S15, there is no surface passivator?

#10: As the for the stability characterization, In Figure 6a, the XRD pattern of initial control film is inconsistent with that in Figure 4c, especially the amount of Pbl₂. In Figure 6g, PCEs tracked under continuous one sun light soaking under open-circuit conditions is not the MPPT, the continuous tracking of power output under the maximum output point should been provided. The description of among the most stable device should also need support of statistic analyzation about the reported works.

Reviewer #3

(Remarks to the Author)

The authors incorporated a multi-site binding ligand of Sb(SU)₂Cl₂ into Pbl₂ layer, which can be transformed into the perovskite after deposition of FAI through 2 step method. This ligand is effective in passivating the various defects, which was well-studied to reveal the interaction between the multi-site binding ligand of Sb(SU)₂Cl₂ and the perovskite. The target device with the ligand showed a high performance (with a PCE of 25%) and a good long-term stability. However, although the defect passivation by the ligand was well-explained, it lacks discussion on the role of the ligand to contribute to forming the perovskite film during the two step process. Thus, I recommend this manuscript to be accepted after several questions and concerns have been adequately addressed.

(1) The authors highlighted a fabrication of perovskite film with a high PCE of 25.03% in the atmosphere by two step method, resulting in the highest values reported for devices fabricated in air. It seems that there is a lack of sufficient understanding regarding the role of ligands during film formation in the two-step process. It seems that more discussion is needed on this matter. To compare this work, please create lists that include the PCEs of recent works that were fabricated in air using the two-step method.

(2) For the Figure 4l, m, residual strain distribution in the perovskite film was discussed, but may be not fully understood yet. The authors need to add more discussion on this issue.

(3) Other than the Pb₀ peak, there is no discernible change between the control and target film's Pb 4f peak positions in the XPS spectrum. This result is not likely to be coincident with the absorption spectrum and the H NMR results, indicating strong coordination interaction of Pb²⁺ and the ligands. Please explain why.

(4) In Figure 3a, FT-IR was performed to investigate the interaction of the ligand with the perovskite. To compare the peaks in the FT-IR, the results of the ligand and the Pbl₂ is necessary.

(5) Similarly, it would be good to add the results of the Pbl₂ w or without treatment of the ligand in the Figure 3b.

(6) In Figure 6e, the Target device showed good thermal stability at a high temperature of 85 oC even if the Spiro-OMeTAD was used as HTL in the presence of dopants with the general condition. Is it right? (Recent publication (<https://doi.org/10.1016/j.joule.2024.10.011>) used different doping condition for improving thermal stability.)

Version 1:

Reviewer comments:

Reviewer #1

(Remarks to the Author)

I am satisfied with the revised manuscript, which has addressed my concerns.

Reviewer #2

(Remarks to the Author)

In the response, the authors have well answered the questions, and supplemented necessary experimental data and discussion. Based on this response, I think this manuscript can be accepted after the following few comments.

(1) In Figure R10 and R11, the binding energy and defect formation energy when the interaction is the on a manner of four-site structure should also be provided.

(2) Only considering the binding energy to claim that the most reasonable form is the four-site structure is inadequate. The match information between the perovskite lattice and molecular length should also be discussed (Tailored Lattice-Matched Carbazole Self-Assembled Molecule for Efficient and Stable Perovskite Solar Cells).

Reviewer #3

(Remarks to the Author)

All inquiries are thoroughly addressed in the revised manuscript. I therefore suggest for the publication of this work in Nature Communications.

- In solar cell reporting summary, the stability of the J-V characteristic for current-voltage characterization is not reflected in this manuscript. Thus, the stabilized power out of the target device (in Figure 4a) needs to be added.

REVIEWER COMMENTS

Reviewer #1 (Remarks to the Author):

This manuscript presents an interesting approach to improving the efficiency and stability of perovskite solar cells (PSCs) by introducing multi-site binding ligands ($\text{Sb}(\text{SU})_2\text{Cl}_3$). The efficiency of the device is impressive for a two-step perovskite solar cell, and the stability is promising. However, there are areas where clarification and additional experimentation would strengthen the manuscript:

#1: The authors emphasize the benefits of the multi-site binding additive ($\text{Sb}(\text{SU})_2\text{Cl}_3$) for two-step fabrication, yet the rationale for its specificity to this method is not entirely clear. Did the authors attempt to incorporate this additive in one-step fabrication? Demonstrating its performance in one-step devices would help confirm whether the additive is inherently advantageous for two-step processing or if its effects are more general.

Our Response: We appreciate the reviewer's insightful comment. We have incorporated this additive in one-step fabrication. After incorporation, the PCE increased from 23.13% to 24.51% (Figures S28 and R1). This result indicates that the additive is also effective in the one-step method, demonstrating its effects are more general. The performance improvement of the device is attributed to the additive-induced enhancement of perovskite crystallization. Please refer to the subsequent comments for detailed evidence and explanations. Please find the revision on Page 14 in the revised manuscript.

Figure R1. J-V curves for one-step methods without and with the additive.

#2: Relatedly, manuscript does not provide a direct explanation of how $\text{Sb}(\text{SU})_2\text{Cl}_3$ specifically addresses challenges unique to the two-step process. For example, how does

the additive influence the interaction between FAI and PbI_2 or mitigate asynchronous crystallization dynamics? The connection between the additive's functionality and the intrinsic characteristics of two-step fabrication should be explicitly discussed.

Our Response: We sincerely thank the reviewer for the constructive suggestions. We have conducted comprehensive multiscale characterizations to elucidate the synergistic mechanism of $\text{Sb}(\text{SU})_2\text{Cl}_3$ in improving the device performance.

1) Regarding how does the additive influences the interaction between FAI and PbI_2 or mitigates asynchronous crystallization dynamics

Crystallization Kinetics (In Situ Absorption Spectroscopy): In situ UV-vis absorption spectroscopy was employed to monitor the crystallization dynamics during the two-step perovskite formation process (Figures 3a, 3b and R2). The crystallization of perovskite in the target film was delayed during thermal annealing, with the transition to α -FAPbI₃ completed at 2.98 s, compared to 1.29 s for the control film. This moderated and orderly crystallization process, attributed to the interaction between $\text{Sb}(\text{TU})_2\text{Cl}_3$ and FAPbI₃, significantly improved film crystallinity and reduced defect density. Please find the revision on Page 11.

Figure R2. In situ UV-Vis absorption spectroscopy for control (a) and target (b) groups

Trap Density and Passivation Effects (SCLC & TSC): SCLC analysis (Figure 4i and Figure R3) demonstrates a 23% reduction in defect density (from $6.76 \times 10^{15} \text{ cm}^{-3}$ to $5.19 \times 10^{15} \text{ cm}^{-3}$) in the coordination complex-treated films, indicative of effective deep-level defect passivation. This trend is corroborated by thermally stimulated current (TSC) spectra, which show a 14% decrease in deep-trap signal intensity ($1.43 \times 10^{15} \text{ cm}^{-3}$ control vs. $1.22 \times 10^{15} \text{ cm}^{-3}$ target, Figure 4h and Figure R4). Please find the revision on Page 17.

Figure R3. Space charge limiting currents (SCLC) are specific to the control group and the control group

Figure R4. Thermally Stimulated Current (TSC) are specific to the control group and the control group

Interfacial Interactions (FTIR Spectroscopy): To probe molecular-level interactions, we performed Fourier-transform infrared (FTIR) spectroscopy on individual components (SU, SbCl₃, and Sb(SU)₂Cl₃) with FAI, PbI₂, and perovskite (PVK) films, respectively (Figures S16 and R5). Our analysis suggests SU interacts with FA⁺ via hydrogen bonds, while SbCl₃ acts as a Lewis acid to enhance electrostatic interactions. In contrast, the Sb(SU)₂Cl₃ exhibits the strongest binding among the components due to its multidentate interaction character. These results highlight the critical role of Cl in stabilizing both precursor and perovskite structures, providing a rational basis for improving film quality and reducing defect states. Please see the revision on Page 10.

Figure R5. Fourier infrared absorption spectroscopy was performed on FAI, PbI₂ and PVK for the control unit and monomer, respectively.

Solution-State Interactions (¹H NMR Spectroscopy): ¹H nuclear magnetic resonance (NMR) spectroscopy was performed to probe interactions between the three additives and FAI, PbI₂, and PVK in solution (Figures S17 and R6), respectively. The interactions may involve: (1) direct Cl coordination with Pb²⁺ to enhance structural stability; (2) SU-mediated hydrogen bond with PbI₂ or FA⁺ to reinforce precursor-cation interactions; and (3) N-H...X (X = I, Se) hydrogen bond to anchor FA⁺ and restrict its mobility, potentially enhancing long-term stability. Our results suggest that the Sb(SU)₂Cl₃ is not a simple mixture of SbCl₃ and SU but a coordinated structure wherein Cl and SU synergistically contribute to form a multidentate interaction network. Compared to the individual additives, the Sb(SU)₂Cl₃ offers several advantages: **a)** Stronger coordination with Pb²⁺, as evidenced by enhanced NMR hydrogen signals in the PbI₂-S system. **b)** Enhanced binding to FA⁺, evidenced by downfield shifts and intensified C=N signals. **c)** Improved precursor solution stability, inferred from the consistent chemical shifts in the PVK-Target system. Please find the revision on Page 10.

Figure R6. Liquid NMR spectroscopy was performed on the control units and monomers of FAI, PbI₂ and PVK, respectively.

2) Regarding the connection between the additive's functionality and the intrinsic characteristics of two-step fabrication:

Compare to one stage method, two-step fabrication has several intrinsic characteristics: 1) The organic salt solution in the second step (such as MAI/FAI) has difficulty fully infiltrating and reacting to form the perovskite phase, resulting in residual unreacted PbI₂. 2) In the two-step method, crystallization occurs in stages (first forming a PbI₂ scaffold, then converting it), which can easily lead to uneven grain sizes or the formation of pinholes.

Based on the above multiscale characterizations, we found that Sb(SU)₂Cl₃ facilitates a multidentate coordination environment that not only improves PbI₂ solubility and strengthens FA⁺ binding but also regulates the crystallization dynamics. This synergy plays a pivotal role in suppressing asynchronous nucleation, enhancing film quality, and eventually improving device performance. Please find the revision on Pages 11 and 17.

#3: One area where the manuscript could be significantly strengthened is through the inclusion of dynamic or in-situ experiments to directly investigate the additive's role in

crystallization kinetics and film formation. Techniques such as in-situ XRD or GIWAXS would offer valuable insights into the phase evolution during the fabrication process, while in-situ PL or TRPL could help track defect dynamics and charge carrier behavior during annealing. Incorporating such experiments would provide robust, real-time evidence to support the proposed mechanisms of improved film quality and enhanced crystallinity induced by $\text{Sb}(\text{SU})_2\text{Cl}_3$, further solidifying the additive's role in achieving these improvements.

Our Response: Following the suggestion, we have conducted additional in situ characterizations to dynamically elucidate the mechanism of $\text{Sb}(\text{SU})_2\text{Cl}_3$.

1) Regarding techniques such as in-situ XRD or GIWAXS would offer valuable insights into the phase evolution during the fabrication process

In Situ GIWAXS Analysis (Figures S23 and R7): The in-situ grazing-incidence wide-angle X-ray scattering (GIWAXS) measurements reveal the α -FAPbI₃ phase during the crystallization process, with a characteristic signal observed at $q \approx 1.0 \text{ \AA}^{-1}$. The target film exhibited α -phase perovskite immediately, while the control film shows it slower. We found that the introduction of $\text{Sb}(\text{TU})_2\text{Cl}_3$ accelerates the nucleation of α -phase perovskite, improving the quality of perovskite films. Please find the revision on Page 11.

Figure R7. In situ GIWAX in the control group (a) and target group (b)

2) Regarding in-situ PL or TRPL could help track defect dynamics and charge carrier behavior during annealing

Crystallization Kinetics (In Situ Absorption Spectroscopy): In situ UV-vis absorption spectroscopy was employed to monitor the crystallization dynamics during the two-step perovskite formation process (Figures 3a, 3b and R2). The crystallization of perovskite in the target film was

delayed during thermal annealing, with the transition to α -FAPbI₃ completed at 2.98 s, compared to 1.29 s for the control film. This moderated and orderly crystallization process, attributed to the interaction between Sb(TU)₂Cl₃ and FAPbI₃, significantly improved film crystallinity and reduced defect density. Please find the revision on Page 11.

Figure R2. In situ UV-Vis absorption spectroscopy for control (a) and target (b) groups

Collectively, these effects work synergistically to accelerate nucleation, promote grain growth, and improve film uniformity and crystallinity. The results demonstrate that the incorporation of Sb(SU)₂Cl₃ provides an effective strategy for modulating the nucleation and growth dynamics of perovskite films, ultimately contributing to enhanced film morphology, reduced defect densities, and improved device stability. Please find the revision in Page 11.

#4: The manuscript could be enhanced by explicitly comparing the stability and efficiency metrics of devices incorporating Sb(SU)₂Cl₃ to other state-of-the-art PSCs fabricated via two-step and one-step methods, especially the MPPT stability at >65°C.

Our Response: We thank the reviewer for the valuable suggestion. We have included comparative tables summarizing the performance of recently reported one-step and two-step processed perovskite devices. These tables highlight that our target device achieves a power conversion efficiency (PCE) of 25.03 %, along with superior thermal stability and MPPT performance. Please find the revision in Tables S2 (Table R1), S3 (Table R2), S4 (Table R3) as well as on Page 18.

Table R1. Summary of high-efficiency (PCE > 22%) thermal stability.

Perovskite	Processing Method	temperature	Relative Humidity (%)	stability	References
FAPbI ₃	Two-step	85	-	T _{97.2} =1400h	This Work
FAPbI ₃	Two-step	85	-	T ₉₉ =1100h	Ref. 1
FAPbI ₃	Two-step	85	85	T ₇₀ =1000h	Ref. 2

FAPbI₃	One-step	85	-	T ₈₀ =700h	Ref. 3
----------	----	---	-----------------------	--------

[1] Yu Zhang, Yanrun Chen, Guilin Liu, Nonalloyed α -phase formamidinium lead triiodide solar cells through iodine intercalation, *Science*, 2025, 387(6731): 284-290.

[2] Lingbo Xiao, Xiaoli Xu, Jie Zhao, High-Temperature Driven Recrystallization for Stable Dopant-Free α -FAPbI₃ Perovskite Solar Cells, *Advanced Science*, 2024, 11(48): 2408684.

[3] Lusheng Liang, Zi-Ang Nan, Yuheng Li, Formation Dynamics of Thermally Stable 1D/3D Perovskite Interfaces for High-Performance Photovoltaics, *Advanced Material*, 2025, 37(8):2413841

Table R2. Summary of high-efficiency (PCE > 22%) MPPT stability.

Perovskite	Processing Method	temperature	stability	References
FAPbI₃	Two-step	25	T_{92.2}=2000h	This Work
FAPbI ₃	One-step	-20	T ₉₀ =1000h	Ref. 1
FA _{0.95} CS _{0.05} PbI ₃	Two-step	30	T ₉₅ =1000h	Ref. 2
FAPbI ₃	Two-step	50	T ₉₄ =1258h	Ref. 3

[1] Muiyang Chen, Tingting Niu, Lingfeng Chao, "Freezing" intermediate phases for efficient and stable FAPbI₃ perovskite solar cells, *Energy Environ. Sci.*, 2024,17, 3375-3383.

[2] Pengju Shi, Yong Ding, Bin Ding, Oriented nucleation in formamidinium perovskite for photovoltaics, *Nature*, 2023, 620: 323-327.

[3] Zijian Huang, Yang Bai, Xudan Huang, Anion- π interactions suppress phase impurities in FAPbI₃ solar cells, *Nature*, 2023, 623: 531-537.

Table R3. A summary of the efficiency of perovskite solar cells prepared by all air method reported in relevant literature and this work.

Perovskite	Champion PCE	Relative humidity	References
FAPbI₃	25.03	20-40	This Work
FAPbI₃	24.70	20	Ref. 1
FAPbI₃	25.70	20	Ref. 2
FAMAPbI₃	24.45	35-45	Ref. 3

[1] Zou Y, Yu WJ, Guo HQ, Li QZ, Li XD, Li L, et al. A crystal capping layer for formation of black-phase FAPbI₃ perovskite in humid air. *Science*. 2024; 385(6705) 161-167.

[2] Yang Y, Huang H, Yan L, Cui P, Lan Z, Sun C, et al. Compatible Soft-Templated Deposition and Surface Molecular Bridge Construction of SnO₂ Enable Air-Fabricated Perovskite Solar Cells with Efficiency Exceeding 25.7%. *Advanced Energy Materials*. 2024; 14(23) 2400416.

[3] Zeng Q, Xiao H, Ma Q, Huang R, Pan Y, Li L, et al. Highly Layer-Oriented PbI₂ Films Enabling All-Air Processed Perovskite Solar Cells. *Advanced Energy Materials*. 2024; 14(32) 2401279.

#5: There is a discrepancy in the description of Figure 6g and 6h. The legend indicates MPPT (maximum power point tracking), but the text describes the condition as open-circuit. This inconsistency should be corrected for clarity and accuracy.

Our Response: We thank the reviewer for pointing out the issue. I apologize for this error. We have

redone the maximum power point tracking (MPPT) test (Figure 5g and 5h, Figure R8), which show that the target device retains 99.98% of its initial PCE after 969 hours of continuous operation. This result further demonstrates the practical potential and long-term operational stability of the device. Please find the revision on Pages 17-18.

Figure R8. Maximum power point tracking (MPPT) measurements for target.

#6: I also recommend shorten the manuscript to 4-5 figures.

Our Response: We thank the reviewer for the suggestion. We have shortened the manuscript to 5 figures, as shown in the revised manuscript.

Reviewer #2 (Remarks to the Author):

In this manuscript, the author incorporated a ligand (a complex of antimony chloride- N,N -dimethyl selenourea, $Sb(SU)_2Cl_3$) with multi-binding-site into PbI_2 precursor in a two-step spin-coating method, thus make it to be an additive of perovskite. It is demonstrated that this multi-site binding ligands can suppress the formation of various defects, enhances the crystallinity of perovskites, and greatly increases the stability of the structure and the ability to resist water, thus attributing to a PCE of 25.03% for air-fabricated PSCs with enhanced stability. Overall, the author proposed a new additive through complexing the antimony chloride and N,N -dimethylselenourea (SU), however the concept of Multi-Site binding with perovskite has been variously reported, and some direct support experiments and analyzation need to supplemented. In addition, the PCE of 25.03% is difficult to support it among the highest values of reported air-fabricated devices. At this stage, it is difficult to recommend the acceptance. Here are the detailed comments.

#1: In the part of introduction: (1) the description of the “three stages in a two-step spin-coating” needs supporting reference, I wonder is there a second stage of involving incorporating water into the reactants, since if we fabricate the perovskite film in glove-box, including the annealing, there is no water. (2) The first paragraph of introduction mainly describes the Asynchronous crystallization, the second paragraph of introduction focus on the undercoordinated Pb^{2+} ions and its corresponding treatment, the logical interaction between these two paragraphs need to be modified and supplemented.

Our Response: The introduction has been systematically revised to clarify the role of water in the crystallization process (Energy Environ. Sci. 18, 1310-1319 (2025); Energy Environ. Sci. 16, 629-640 (2023); Small 20, 2404044 (2024); Science 383, 524-531 (2024); Nat. Commun. 13, 4891 (2022); Adv. Funct. Mater. 34, 2403690 (2024).). The logical interaction between the first paragraph and the second paragraph is also improved to reach a better flow. Please find the revision on Page 3.

#2: There experimental data proved that the interaction between $\text{Sb}(\text{SU})_2\text{Cl}_3$ and perovskite can also be Pb-N and hydrogen bond, why theses bindings have not been included in the DFT calculation?

Our Response: Thanks for the suggestion. We have performed additional calculations on the interactions between $\text{Sb}(\text{SU})_2\text{Cl}_3$ and perovskite in form of interfacial Pb-N and hydrogen bonds. The atomic structure and the adsorption energies are shown in Figure S6 (Figure R10). The structure with hydrogen bonds exhibits an adsorption energy of only -0.11 eV, since the hydrogen bond interaction is much weaker than the chemical bond interaction.

The structure with interfacial Pb-N bond exhibits an adsorption energy of -0.39 eV, which is greater than the one with hydrogen bond but still much lower than the one with interaction between Cl/Se and Pb atoms. This is because the N atom is surrounded by H atoms and close to a saturated state. Moreover, the H atoms impede an effective interaction of the N atom with the perovskite surface. As a result, the surface Pb atoms are more favorable to bond with Cl or Se atoms in the molecule, yielding a stronger adsorption and improving the stability. Thus, we focus our calculations on the structures with Cl-Pb and Se-Pb bonds. Please find the revision on Page 9.

Figure R10. Side views of the hydrogen bond interactions (a) and the N-Pb interactions (b) between the molecule and the perovskite surface. (c) The adsorption energies of six structures with distinct interfacial bonding between the molecule and the perovskite surface.

#3: Considering there are various binding approaches (one site, two site four site, maybe three site, Pb-N, Pb-Se, Pb-Cl and hydrogen bond), it would be better and meaningful to provide a discussion about the practical interaction method between perovskite and additives, or which binding approach the author think is most reasonable and possesses more possibilities.

Our Response: First, according to our calculations of the interaction between the $\text{Sb}(\text{SU})_2\text{Cl}_3$ molecule and the perovskite surface in the manuscript and the reply to Comment 2, the adsorption energy of the four-site structure is the lowest. This indicates that the molecule is most likely to exist in this form on perovskite. Meanwhile, the four-site configuration exhibits the most pronounced charge transfer across the interface, which is coherent with the enhanced interfacial coupling and improved passivation of the perovskite surface.

To further demonstrate this passivation effect, we focused on the I vacancy defect (V_I), the most prevalent in perovskite. We calculated the defect formation energies of passivated perovskite surface with the $\text{Sb}(\text{SU})_2\text{Cl}_3$ binding to different sites (Figures S8 and R11). The results show that the four-site structure delivers the highest V_I formation energy, thereby holding the propensity of suppressing the defect formation and enhancing the perovskite stability. Please find the revision on Page 9.

Figure R11. Defect formation energies of V_I for the pristine perovskite and after the $Sb(SU)_2Cl_3$ molecule modified at six different sites.

In addition, to further illustrate the role of multiple sites, we conducted separate calculations on the two component groups of $Sb(SU)_2Cl_3$, *i.e.*, the SU and $SbCl_3$. Their optimal adsorption structures on the perovskite surface are shown in Figure S9 (Figure R12). The SU group interacts with the perovskite surface via a Se-Pb bond, with an adsorption energy of -0.65 eV (Figure R12c). The $SbCl_3$ group is bound on the perovskite surface with two Cl-Pb bonds, with an adsorption energy of -1.36 eV. These results further rationalize our four-site structure. The complexation of the SU and the $SbCl_3$ to form the $Sb(SU)_2Cl_3$ molecule further increases the adsorption sites and, thus, the adsorption energy, endowing the corresponding adsorption structure with enhanced stability.

We have also calculated the defect formation energies of V_I for the SU- and the $SbCl_3$ -adsorbed perovskite surfaces (Figure R12d). The defect formation energy increases in both cases compared to that for the pristine perovskite, but remains considerably lower than that for the $Sb(SU)_2Cl_3$ -adsorbed surface. The increased adsorption energy and defect formation energy of V_I once again confirm that the four-site adsorption configuration of the $Sb(SU)_2Cl_3$ molecule is more reasonable and of practical relevance. Please find the revision on Page 9.

Figure R12. Side views of the adsorption structures of the SU molecule (a) and the SbCl₃ molecule (b) on the perovskite surface. (c) The adsorption energies of the SU molecule, the SbCl₃ molecule, and the Sb(SU)₂Cl₃ molecule on the perovskite surface. (d) Defect formation energies of V₁ for the perovskite surface after being modified by the SU molecule, the SbCl₃ molecule, and the Sb(SU)₂Cl₃ molecule.

#4: Considering the core idea of this work is enhanced coordination interaction, the direct comparative experiment is suggested to be supplemented to prove that the complexing can enhance the binding energy.

Our Response: We thank the reviewer for the valuable suggestion. We have supplemented the direct comparative experiment to prove that the complexing can enhance the binding energy.

The peak intensity corresponding to Pb⁰ reduced after incorporating multi-site binding ligands (Sb(SU)₂Cl₃), indicating that the residual PbI₂ is largely reduced. The binding energy of the Pb 4f orbital is increased by 0.25 eV after incorporating multi-site binding ligands (Sb(SU)₂Cl₃). The chemical shifts toward higher binding energies would result from the strong chemical interaction between the Sb(SU)₂Cl₃ and positively charged under-coordinated Pb²⁺ ions. Please find the revision in Figure 3e-f (Figure R13) and on Page 12.

Figure R13. High-resolution Pb 4f XPS peaks of the control and target perovskite film.

#5: In **Figure 3**, the main data focus on the chemical states changes of FA⁺ (Supporting the interaction between FAI and additives), however, the DFT results in **Figure 2** reveal that the all four binding sites focus on the Pb²⁺. The experiments that can prove the enhanced coordination between additive with Pb²⁺ is suggested to be supplemented. In addition, the author mentioned that “NMR spectroscopy further highlighted significant effects of the complex on the chemical environments of FA⁺ and Pb²⁺ ions.”, however in the following discussion there no discussion of the chemical environments of Pb²⁺ ions. The peak position and peak shift are suggested to be marked in **Figure 3a**

Our Response: We sincerely thank the reviewer for the insightful suggestions.

- 1) The experiments that can prove the enhanced coordination between additive and Pb²⁺ have been supplemented.

Interfacial Interactions (FTIR Spectroscopy): To probe molecular-level interactions, we performed Fourier-transform infrared (FTIR) spectroscopy on individual components (SU, SbCl₃, and Sb(SU)₂Cl₃) with FAI, PbI₂, and perovskite (PVK) films, respectively (Figures S16 and R5). Our analysis suggests SU interacts with FA⁺ via hydrogen bonds, while SbCl₃ acts as a Lewis acid to

enhance electrostatic interactions. In contrast, the $\text{Sb}(\text{SU})_2\text{Cl}_3$ exhibits the strongest binding among the components due to its multidentate interaction character. These results highlight the critical role of Cl in stabilizing both precursor and perovskite structures, providing a rational basis for improving film quality and reducing defect states. Please see the revision on Page 10.

Figure R5. Fourier infrared absorption spectroscopy was performed on FAI, PbI_2 and PVK for the control unit and monomer, respectively.

2) We have discussed the chemical environments of Pb^{2+} ions by NMR spectroscopy

Solution-State Interactions (^1H NMR Spectroscopy): ^1H nuclear magnetic resonance (NMR) spectroscopy was performed to probe interactions between the three additives and FAI, PbI_2 , and PVK in solution (Figure S17 and Figure R6). In the PbI_2 system, no distinct hydrogen signals were observed in the 7-9 ppm region for PbI_2 and $\text{PbI}_2\text{-SbCl}_3$, suggesting SbCl_3 primarily coordinates with Pb^{2+} via Cl without perturbing the proton environment. In contrast, $\text{PbI}_2\text{-SU}$ and $\text{PbI}_2\text{-Target}$ displayed significant signal enhancements, indicating interactions through selenourea (SU) or N-H bonds. The stronger signals in the $\text{PbI}_2\text{-Target}$ system suggest more stable coordination networks involving both Cl and SU groups.

These interactions may involve: (1) direct Cl coordination with Pb^{2+} to enhance structural stability; (2) SU-mediated hydrogen bonding with PbI_2 or FA^+ to reinforce precursor-cation

interactions; and (3) N-H...X (X = I, Se) hydrogen bonding to anchor FA⁺ and restrict its mobility, potentially enhancing long-term stability. These results suggest that the Sb(SU)₂Cl₃ is not a simple mixture of SbCl₃ and SU but a coordinated structure wherein Cl and SU can synergistically contribute to a multidentate interaction network. Compared to the individual additives, the Sb(SU)₂Cl₃ offers several advantages: **a)** Stronger coordination with Pb²⁺, as evidenced by enhanced NMR hydrogen signals in the PbI₂-S system. **b)** Enhanced binding to FA⁺, demonstrated by downfield shifts and intensified C=N signals. **c)** Improved precursor solution stability, inferred from the consistent chemical shifts in the PVK-Target system.

Collectively, these multiscale characterizations reveal that Sb(SU)₂Cl₃ facilitates a multidentate coordination environment that not only improves PbI₂ solubility and strengthens FA⁺ binding but also regulates crystallization dynamics. This synergy plays a pivotal role in suppressing asynchronous nucleation, enhancing film quality, and improving device stability. Please find the revision in Figure S17 (Figure R6) and on Page 10.

Figure R6. Liquid NMR spectroscopy was performed on the control units and monomers of FAI, PbI₂ and PVK, respectively.

- 3) The peak position and peak shift have been marked in Figure 2g, please find the revision in Figure 2g (Figure R14) and on Page 8.

Figure R 14. FTIR spectra of FAI and perovskite without and with $\text{Sb}(\text{SU})_2\text{Cl}_3$. The blue bands denote the N-H stretching peaks.

#6: In **Figure 4a-4d**, the denser morphology and lower PbI_2 peaks can not support the conclusion of improved crystallinity, the FWHM should be analyzed. What the reason for the optimized crystallization? especially considering the ambient air condition. The in-situ measurement, is suggested to be performed to reveal the influence of additive on perovskite crystallization.

Our Response: We gratefully acknowledge the reviewer's constructive suggestion.

1) The FWHM has been analyzed.

The full width at half maximum (FWHM) was reduced to some extent after incorporating the $\text{Sb}(\text{SU})_2\text{Cl}_3$, suggesting an improvement in the crystallinity of the material (Figures S20 and R15). Please see the revision in Page 11. In addition, we also carried out the in situ UV-vis absorption measurements and in-situ grazing-incidence wide-angle X-ray scattering (GIWAXS) measurements to prove the improved crystallinity. Please refer to the following response.

Figure R15. XRD of the control and target samples.

2) What the reason for the optimized crystallization? especially considering the ambient air condition.

According to the analysis in the comment 3, SU interacts with FA^+ through hydrogen bonding, while Sb^{3+} , acting as a Lewis acid, enhances electrostatic interactions. Among all systems, $\text{Sb}(\text{SU})_2\text{Cl}_3$ exhibits the strongest binding affinity due to its multidentate coordination capability. These findings highlight the critical role of Cl^- in stabilizing both the precursor and perovskite structures, providing a rational basis for improving film quality and reducing defect states. The results indicate that $\text{Sb}(\text{SU})_2\text{Cl}_3$ is not a simple mixture of Sb and SU, but rather a coordinated complex in which Cl^- and SU synergistically form a multidentate interaction network. Compared to individual additives, $\text{Sb}(\text{SU})_2\text{Cl}_3$ offers several advantages: (a) stronger coordination with Pb^{2+} , as evidenced by the enhanced NMR hydrogen signal in the $\text{PbI}_2\text{-S}$ system; (b) improved binding with FA^+ , supported by downfield chemical shifts and intensified $\text{C}=\text{N}$ signals; and (c) enhanced stability of the precursor solution. When the perovskite film is fabricated in air, these interactions may influence the contact between the perovskite precursors and moisture, thereby affecting the crystallization of the perovskite. Please see Figures S16 and S17.

3) Furthermore, to explain the reason for the optimized crystallization, the in-situ measurement has also been performed to reveal the influence of additive on perovskite crystallization.

Crystallization Kinetics (In Situ Absorption Spectroscopy): In situ UV-vis absorption

spectroscopy was employed to monitor the crystallization dynamics during the two-step perovskite formation process (Figures 3a, 3b and R2). The crystallization of perovskite in the target film was delayed during thermal annealing, with the transition to α -FAPbI₃ completed at 2.98 s, compared to 1.29 s for the control film. This moderated and orderly crystallization process, attributed to the interaction between Sb(TU)₂Cl₃ and FAPbI₃, significantly improved film crystallinity and reduced defect density. Please find the revision on Page 11.

Figure R2. In situ UV-Vis absorption spectroscopy for control (a) and target (b) groups

In Situ GIWAXS Analysis (Figures S23 and R7): The in-situ grazing-incidence wide-angle X-ray scattering (GIWAXS) measurements reveal the α -FAPbI₃ phase during the crystallization process, with a characteristic signal observed at $q \approx 1.0 \text{ \AA}^{-1}$. The target film exhibited α -phase perovskite immediately, while the control film shows it slower. We found that the introduction of Sb(TU)₂Cl₃ accelerates the nucleation of α -phase perovskite, improving the quality of perovskite films. Please find the revision on Page 11.

Figure R7. In situ GIWAX in the control group (a) and target group (b)

#7: Considering the main effects of the additive is reducing the defects in perovskite film, the direct experiments to demonstrate that the reduced defects should be supplemented, such as SCLC, thermal admittance spectroscopy....

Our Response: We appreciate the reviewer's valuable comment.

Because thermal admittance spectroscopy is not available in our lab, instead, we performed complementary space-charge-limited current (SCLC) and thermally stimulated current (TSC) measurements.

SCLC analysis (Figures 4i and R3) reveals a 23% reduction in defect density (from 6.76×10^{15} to $5.19 \times 10^{15} \text{ cm}^{-3}$) in the coordination complex-treated films, indicative of effective deep-level defect passivation. This trend is corroborated by measured TSC spectra, which show a 14% decrease in deep-trap signal intensity ($1.43 \times 10^{15} \text{ cm}^{-3}$ control vs. $1.22 \times 10^{15} \text{ cm}^{-3}$ target, Figures 4h and R4). Please find the revision on Page 17.

Figure R3. Space charge limiting currents (SCLC) are specific to the control group and the control group

Figure R4. Thermally Stimulated Current (TSC) are specific to the control group and the control group

#8: In **Figure 5**, The parameter statistics of control PSCs under reverse and forward scan should be provided. Photovoltaic parameter statistics of multiple devices need to be supplemented. In addition, among the highest efficiencies is a too broad description, if the author wants to evaluate this efficiency in this way, the efficiency statistic of the reported works should be provided to make a direct comparison. Besides, the certificated PCE should also be provided.

Our Response: We thank the reviewer for this critical suggestion.

Figures S27 and R16 shows the J-V hysteresis of the control and target cells. After incorporating $\text{Sb}(\text{SU})_2\text{Cl}_3$, the hysteresis is reduced. Please find the revision on Page 14.

Figure R16. Current density-voltage curves of the a) control and b) target PSCs measured in the reverse and forward mode.

To rigorously validate device performance, we have conducted statistical analysis across 15 independent control and target devices; the obtained box-plot distributions shown in Figures S29 and R17 demonstrates exceptional reproducibility.

Figure R17. Statistical analysis across 15 independent control and target devices.

We have provided the efficiency statistic of the reported works to make a direct comparison in Table R3.

Table R3. A summary of the efficiency of perovskite solar cells prepared by all air method reported in relevant literature and this work.

Perovskite	Champion PCE	Relative humidity	References
FAPbI₃	25.03	20-40	This Work
FAPbI ₃	24.70	20	Ref. 1
FAPbI ₃	25.70	20	Ref. 2
FAMAPbI ₃	24.45	35-45	Ref. 3

[1] Zou Y, Yu WJ, Guo HQ, Li QZ, Li XD, Li L, et al. A crystal capping layer for formation of black-phase FAPbI₃ perovskite in humid air. *Science*. 2024; 385(6705) 161-167.

[2] Yang Y, Huang H, Yan L, Cui P, Lan Z, Sun C, et al. Compatible Soft-Templated Deposition and Surface Molecular Bridge Construction of SnO₂ Enable Air-Fabricated Perovskite Solar Cells with Efficiency Exceeding 25.7%. *Advanced Energy Materials*. 2024; 14(23) 2400416.

[3] Zeng Q, Xiao H, Ma Q, Huang R, Pan Y, Li L, et al. Highly Layer-Oriented PbI₂ Films Enabling All-Air Processed Perovskite Solar Cells. *Advanced Energy Materials*. 2024; 14(32) 2401279.

Furthermore, we have certified the efficiency of the solar cells last year. The third-party certification under AM 1.5G illumination (25°C, 50% RH) by Shanghai Institute of Microsystem and Information Technology Chinese Academy of Sciences (SIMIT) gave a calibrated PCE of 24.29 and

24.34% in the forward and reverse scan mode, respectively. Please find it in Figures S26 and R18.

Report No. 24TR012002

====Measurement Results====

	Forward Scan (Isc to Voc)	Reverse Scan (Voc to Isc)
Area	4.40 mm ²	
Isc	1.079 mA	1.079 mA
Voc	1.183 V	1.184 V
Pmax	1.069 mW	1.071 mW
Ipm	1.043 mA	1.046 mA
Vpm	1.025 V	1.024 V
FF	83.72 %	83.84 %
Eff	24.29 %	24.34 %

- Active area was provided by client.
- Test results listed in this measurement report refer exclusively to the mentioned measured sample.
- The results apply only at the time of the test, and do not imply future performance.

Fig.1 I-V curves of the measured sample

-----End of Report-----

Figure R18. Certificated results from an accredited photovoltaic certification laboratory (Shanghai Institute of Microsystem and Information Technology Chinese Academy of Sciences (SIMIT)).

#9: Is there any surface passivation? In **Figure S15**, there is no surface passivator?

Our Response: Yes, we used the corresponding passivation in the actual device preparation process. Now, we have labelled the PEAI passivation layer in Figure S25 (Figure R19) in the revised manuscript.

Figure R19. Device structure of the perovskite solar cells with and without Sb(SU)₂Cl₃.

#10: As the for the stability characterization, In **Figure 6a**, the XRD pattern of initial control film is inconsistent with that in **Figure 4c**, especially the amount of Pbl₂. In **Figure 6g**, PCEs tracked under continuous one sun light soaking under open-circuit conditions is not the MPPT, the continuous tracking of power output under the maximum output point should been provided. The description of among the most stable device should also need support of statistic analyzation about the reported works.

Our Response: We thank the reviewer for the valuable suggestion.

In the revised manuscript, we repeated the experiment to harmonize the initial conditions and re-measured the XRD of the control films for supporting the desired stability (Figure 5a-b (Figure R20)).

Figure R20. a) and b) XRD patterns of control and target perovskite films aged in ambient condition (70% humidity) and at 110 °C. The samples were prepared in the atmosphere (20-40% humidity) and at room temperature.

Additionally, we have performed maximum power point tracking (MPPT) measurements (Figures 5g and R21), which show that the target cell maintained 99.98% of its initial power conversion efficiency (PCE) after 969 hours of continuous operation under one-sun illumination. This result further demonstrates the practical potential and long-term operational stability of the device. Please find the revision on Page 18.

Figure R21. maximum power point tracking (MPPT) measurements for target.

We have included comparative tables summarizing the performance of recently reported

perovskite solar cells. These tables highlight that our target device achieves a power conversion efficiency (PCE) of 25.03%, along with superior thermal stability and MPPT performance, thus supporting the significance of our findings. Please find the revision in Table S2 (Table R1), Table S3 (Table R2) and on Page 18.

Table R1. Summary of high-efficiency (PCE > 22%) thermal stability.

Perovskite	Processing Method	temperature	Relative Humidity (%)	stability	References
FAPbI ₃	Two-step	85	-	T _{97.2} =1400h	This Work
FAPbI ₃	Two-step	85	-	T ₉₉ =1100h	Ref. 1
FAPbI ₃	Two-step	85	85	T ₇₀ =1000h	Ref. 2
FAPbI ₃	One-step	85	-	T ₈₀ =700h	Ref. 3

[1] Yu Zhang, Yanrun Chen, Guilin Liu, Nonalloyed α -phase formamidinium lead triiodide solar cells through iodine intercalation, *Science*, 2025, 387(6731): 284-290.

[2] Lingbo Xiao, Xiaoli Xu, Jie Zhao, High-Temperature Driven Recrystallization for Stable Dopant-Free α -FAPbI₃ Perovskite Solar Cells, *Advanced Science*, 2024, 11(48): 2408684.

[3] Lusheng Liang, Zi-Ang Nan, Yuheng Li, Formation Dynamics of Thermally Stable 1D/3D Perovskite Interfaces for High-Performance Photovoltaics, *Advanced Material*, 2025, 37(8):2413841

Table R2. Summary of high-efficiency (PCE > 22%) MPPT stability.

Perovskite	Processing Method	temperature	stability	References
FAPbI ₃	Two-step	25	T _{92.2} =2000h	This Work
FAPbI ₃	One-step	-20	T ₉₀ =1000h	Ref. 1
FA _{0.95} CS _{0.05} PbI ₃	Two-step	30	T ₉₅ =1000h	Ref. 2
FAPbI ₃	Two-step	50	T ₉₄ =1258h	Ref. 3

[1] Muiyang Chen, Tingting Niu, Lingfeng Chao, "Freezing" intermediate phases for efficient and stable FAPbI₃ perovskite solar cells, *Energy Environ. Sci.*, 2024,17, 3375-3383.

[2] Pengju Shi, Yong Ding, Bin Ding, Oriented nucleation in formamidinium perovskite for photovoltaics, *Nature*, 2023, 620: 323-327.

[3] Zijian Huang, Yang Bai, Xudan Huang, Anion- π interactions suppress phase impurities in FAPbI₃ solar cells, *Nature*, 2023, 623: 531-537.

Reviewer #3 (Remarks to the Author):

The authors incorporated a multi-site binding ligand of Sb(SU)₂Cl₂ into Pbl₂ layer, which can be transformed into the perovskite after deposition of FAI through 2 step method. This ligand is effective in passivating the various defects, which was well-studied to reveal the interaction between the multi-site binding ligand of Sb(SU)₂Cl₂ and the perovskite. The target device with the ligand showed a high performance (with a PCE of 25%) and a good long-

term stability. However, although the defect passivation by the ligand was well-explained, it lacks discussion on the role of the ligand to contribute to forming the perovskite film during the two step process. Thus, I recommend this manuscript to be accepted after several questions and concerns have been adequately addressed.

#1: The authors highlighted a fabrication of perovskite film with a high PCE of 25.03% in the atmosphere by two step method, resulting in the highest values reported for devices fabricated in air. It seems that there is a lack of sufficient understanding regarding the role of ligands during film formation in the two-step process. It seems that more discussion is needed on this matter. To compare this work, please create lists that include the PCEs of recent works that were fabricated in air using the two-step method.

Our Response: We sincerely thank the reviewer for the insightful suggestions. In response, we conducted comprehensive multiscale characterizations to elucidate the synergistic mechanism of $\text{Sb}(\text{SU})_2\text{Cl}_3$ in forming the perovskite films.

Crystallization Kinetics (In Situ Absorption Spectroscopy): In situ UV-vis absorption spectroscopy was employed to monitor the crystallization dynamics during the two-step perovskite formation process (Figures 3a, 3b and R2). The crystallization of perovskite in the target film was delayed during thermal annealing, with the transition to α -FAPbI₃ completed at 2.98 s, compared to 1.29 s for the control film. This moderated and orderly crystallization process, attributed to the interaction between $\text{Sb}(\text{TU})_2\text{Cl}_3$ and FAPbI₃, significantly improved film crystallinity and reduced defect density. Please find the revision on Page 11.

Figure R2. In situ UV-Vis absorption spectroscopy for control (a) and target (b) groups

In Situ GIWAXS Analysis (Figures S23 and R7): The in-situ grazing-incidence wide-angle X-ray scattering (GIWAXS) measurements reveal the α -FAPbI₃ phase during the crystallization process, with a characteristic signal observed at $q \approx 1.0 \text{ \AA}^{-1}$. The target film exhibited α -phase perovskite immediately, while the control film shows it slower. We found that the introduction of Sb(TU)₂Cl₃ accelerates the nucleation of α -phase perovskite, improving the quality of perovskite films. Please find the revision on Page 11.

Figure R7. In situ GIWAX in the control group (a) and target group (b)

We have included comparative tables summarizing the performance of recently reported perovskite solar cells. These tables highlight that our target device achieves a power conversion efficiency (PCE) of 25.03%, along with superior thermal stability and MPPT performance, thereby supporting the significance of our findings. Please find the revision in Table S2 (Table R1), Table S3 (Table R2), Table S4 (Table R3), and on Page 18.

Table R1. Summary of high-efficiency (PCE > 22%) thermal stability.

Perovskite	Processing Method	temperature	Relative Humidity (%)	stability	References
FAPbI ₃	Two-step	85	-	T _{97.2} =1400h	This Work
FAPbI ₃	Two-step	85	-	T ₉₉ =1100h	Ref. 1
FAPbI ₃	Two-step	85	85	T ₇₀ =1000h	Ref. 2
FAPbI ₃	One-step	85	-	T ₈₀ =700h	Ref. 3

[1] Yu Zhang, Yanrun Chen, Guilin Liu, Nonalloyed α -phase formamidinium lead triiodide solar cells through iodine intercalation, *Science*, 2025, 387(6731): 284-290.

[2] Lingbo Xiao, Xiaoli Xu, Jie Zhao, High-Temperature Driven Recrystallization for Stable Dopant-Free α -FAPbI₃ Perovskite Solar Cells, *Advanced Science*, 2024, 11(48): 2408684.

[3] Lusheng Liang, Zi-Ang Nan, Yuheng Li, Formation Dynamics of Thermally Stable 1D/3D Perovskite Interfaces for High-Performance Photovoltaics, *Advanced Material*, 2025, 37(8):2413841

Table R2. Summary of high-efficiency (PCE > 22%) MPPT stability.

Perovskite	Processing Method	temperature	stability	References
FAPbI ₃	Two-step	25	T _{92.2} =2000h	This Work
FAPbI ₃	One-step	-20	T ₉₀ =1000h	Ref. 1
FA _{0.95} CS _{0.05} PbI ₃	Two-step	30	T ₉₅ =1000h	Ref. 2
FAPbI ₃	Two-step	50	T ₉₄ =1258h	Ref. 3

[1] Muiyang Chen, Tingting Niu, Lingfeng Chao, "Freezing" intermediate phases for efficient and stable FAPbI₃ perovskite solar cells, *Energy Environ. Sci.*, 2024,17, 3375-3383.

[2] Pengju Shi, Yong Ding, Bin Ding, Oriented nucleation in formamidinium perovskite for photovoltaics, *Nature*, 2023, 620: 323-327.

[3] Zijian Huang, Yang Bai, Xudan Huang, Anion- π interactions suppress phase impurities in FAPbI₃ solar cells, *Nature*, 2023, 623: 531-537.

Table R3. A summary of the efficiency of perovskite solar cells prepared by all air method reported in relevant literature and this work.

Perovskite	Champion PCE	Relative humidity	References
FAPbI ₃	25.03	20-40	This Work
FAPbI ₃	24.70	20	Ref. 1
FAPbI ₃	25.70	20	Ref. 2
FAMAPbI ₃	24.45	35-45	Ref. 3

[1] Zou Y, Yu WJ, Guo HQ, Li QZ, Li XD, Li L, et al. A crystal capping layer for formation of black-phase FAPbI₃ perovskite in humid air. *Science*. 2024; 385(6705) 161-167.

[2] Yang Y, Huang H, Yan L, Cui P, Lan Z, Sun C, et al. Compatible Soft-Templated Deposition and Surface Molecular Bridge Construction of SnO₂ Enable Air-Fabricated Perovskite Solar Cells with Efficiency Exceeding 25.7%. *Advanced Energy Materials*. 2024; 14(23) 2400416.

[3] Zeng Q, Xiao H, Ma Q, Huang R, Pan Y, Li L, et al. Highly Layer-Oriented PbI₂ Films Enabling All-Air Processed Perovskite Solar Cells. *Advanced Energy Materials*. 2024; 14(32) 2401279.

#2: For the **Figure 4l, m**, residual strain distribution in the perovskite film was discussed, but may be not fully understood yet. The authors need to add more discussion on this issue.

Our Response: The corresponding out-of-plane line cuts of the GIWAXS images as a function of incidence angles can directly reflect the d-spacing difference from the surface to bottom. To visually reflect residual strain distribution in the perovskite film, grazing-incidence X-ray diffraction (GIXRD) analysis was carried out, revealing a depth-dependent variation in residual stress and microstrain within the (012) plane of the perovskite films. After incorporating the Sb(SU)₂Cl₃, the absolute value of stress from 16.3 MPa (Control) to 8.6 MPa (Target), indicating that the incorporation of the Sb(SU)₂Cl₃ released the residual stress of the perovskite film to some extent. Please find the revision

in Figures 3c, 3d, S24 and R22 and on Pages 11 and 12.

Figure R22. Grazing-incidence X-ray diffraction (GIXRD) profiles of perovskite thin films (012) crystallographic plane.

#3: Other than the Pb^0 peak, there is no discernible change between the control and target film's Pb 4f peak positions in the XPS spectrum. This result is not likely to be coincident with the absorption spectrum and the H NMR results, indicating strong coordination interaction of Pb^{2+} and the ligands. Please explain why.

Our Response: The XPS test depth (~5 nm) limitation may have caused the Pb 4f peak shift to be less pronounced; thus, we further re-tested the XPS for interpretation. The peak intensity corresponding to Pb^0 reduced after incorporating multi-site binding ligands ($Sb(SU)_2Cl_3$), indicating that the residual PbI_2 is largely reduced. The binding energy of the Pb 4f orbital is increased by 0.25 eV after incorporating multi-site binding ligands ($Sb(SU)_2Cl_3$). The chemical shifts toward higher binding energies would result from the strong chemical interaction between the $Sb(SU)_2Cl_3$ and positively charged under-coordinated Pb^{2+} ions. Please find the revision in Figures 3e-f (Figure R13) and on Page 12.

Figure R13. High-resolution Pb 4f XPS peaks of the control and target perovskite film.

#4: In **Figure 3a**, FT-IR was performed to investigate the interaction of the ligand with the perovskite. To compare the peaks in the FT-IR, the results of the ligand and the PbI_2 is necessary.

Our Response: We sincerely thank the reviewer for the insightful suggestion. We have added the FTIR comparison of PbI_2 with and without $\text{Sb}(\text{SU})_2\text{Cl}_3$. Please find the revision in Figure S16 (Figure R5) and on Page 10.

Comparative analysis of the interactions between SbCl_3 , SU, and $\text{Sb}(\text{SU})_2\text{Cl}_3$ with the perovskite precursor reveals that this complex exhibits the strongest binding affinity among all systems studied. Notably, this complex is not merely a physical mixture of SbCl_3 and SU; rather, it forms a much more stable PbI_2 coordination environment through a synergy between halide ions and SU group. This structural arrangement simultaneously enhances the binding strength of $\text{Sb}(\text{SU})_2\text{Cl}_3$ to FA^+ ions.

Figure R5. Fourier infrared absorption spectroscopy was performed on FAI, PbI₂ and PVK for the control unit and monomer, respectively.

#5: Similarly, it would be good to add the results of the PbI₂ w or without treatment of the ligand in **Figure 3b**.

Our Response: We have added the results of the PbI₂ with or without treatment of the ligand in Figure 2h. The UV-vis absorption spectra of PbI₂ and its complex with Sb(SU)₂Cl₃ in DMF reveal significantly enhanced absorbance in the 370-440 nm region after treatment, accompanied by a slight redshift in the absorption edge. This indicates strong coordination interactions between the multidentate the Sb(SU)₂Cl₃ ligand and Pb²⁺ ions. These interactions perturb the Pb-I coordination environment and lead to new charge transfer process, confirming the formation of a stable complex in solution, which benefits subsequent perovskite film crystallization and quality improvement. Please find the revision in Figure 2h and Figure R23.

Figure R23: UV-vis absorption spectra of PbI_2 and the $\text{PbI}_2\text{-Sb(SU)}_2\text{Cl}_3$ complex in DMF solution.

#6: In **Figure 6e**, the Target device showed good thermal stability at a high temperature of 85 °C even if the Spiro-OMeTAD was used as HTL in the presence of dopants with the general condition. Is it right? (Recent publication (<https://doi.org/10.1016/j.joule.2024.10.011>) used different doping condition for improving thermal stability.)

Our Response: We thank the reviewers for their suggestions. We have cited the mentioned reference. To better reflect the intrinsic stability of the perovskite itself, we used poly[bis(4-phenyl)(2,4,6-trimethylphenyl)amine (PTAA) as the HTL to improve the thermal stability, with the details described on Pages 17, and 21 in the revision.

Reviewer #1 (Remarks to the Author):

I am satisfied with the revised manuscript, which has addressed my concerns.

Response: We thank the reviewer for the highly positive evaluation.

Reviewer #2 (Remarks to the Author):

In the response, the authors have well answered the questions, and supplemented necessary experimental data and discussion. Based on this response, I think this manuscript can be accepted after the following few comments.

Response: We thank the reviewer for the highly positive evaluation and offering useful suggestions to improve the manuscript.

(1) In Figure R10 and R11, the binding energy and defect formation energy when the interaction is the on a manner of four-site structure should also be provided.

Response: Thanks for the comment. We apologize for our previous different expressions of the four-site structure in the manuscript and supporting information. In the previous Figure R10 and R11 (Figure S8 and S9), the “2Se + Cl” means the four-site structure, which might have caused ambiguity. Now we have uniformly changed the names of all the four-site structures to “2Se + 2Cl” (Figure R1 and R2).

Figure R1. Side views of the hydrogen bond interactions (a) and the N-Pb interactions (b) between the molecule and the perovskite surface. (c) The adsorption energies of six different interactions between the molecule and the perovskite surface.

Figure R2. Defect formation energies of V_I for the pristine perovskite and after the $Sb(SU)_2Cl_3$ molecule modified at six different sites.

(2) Only considering the binding energy to claim that the most reasonable form is the four-site structure is inadequate. The match information between the perovskite lattice and molecular length should also be discussed (Tailored Lattice-Matched Carbazole Self-Assembled Molecule for Efficient and Stable Perovskite Solar Cells).

Response: Thanks for the valuable suggestion. As shown in Figure R3 (Figure S5), the lattice of $FAPbI_3$ is a square with a diagonal of 8.9 Å, while the Se and Cl atoms of the $Sb(SU)_2Cl_3$ can precisely form a square with a diagonal of 6.1 Å. Although this square is slightly smaller than the lattice of $FAPbI_3$, due to the appropriate atomic radii and abundant outer electrons of Se, Cl, and Pb atoms, the Pb-Cl bonds and Pb-Se bonds can be formed. Therefore, the Se-Cl square can nearly perfectly match an $FAPbI_3$ lattice and thereby achieve passivation, which is consistent with the previous research [1].

Figure R3. The top view of the $Sb(SU)_2Cl_3$ treated $FAPbI_3$ surface.

[1] Tailored Lattice-Matched Carbazole Self-Assembled Molecule for Efficient and Stable Perovskite Solar Cells

Reviewer #3 (Remarks to the Author):

All inquiries are thoroughly addressed in the revised manuscript. I therefore suggest for the publication of this work in Nature Communications.

Response: We thank the reviewer for the highly positive evaluation.

- In solar cell reporting summary, the stability of the J-V characteristic for current-voltage characterization is not reflected in this manuscript. Thus, the stabilized power out of the target device (in Figure 4a) needs to be added.

Response: Thanks for the valuable suggestion. We have added the stabilized power out of the target device in Figure 4a.